# Geoclidean: Few-Shot Generalization in Euclidean Geometry

**Joy Hsu**
Computer Science
Stanford University
joycj@stanford.edu

**Jiajun Wu**
Computer Science
Stanford University
jiajunwu@cs.stanford.edu

**Noah D. Goodman**
Psychology and Computer Science
Stanford University
ngoodman@stanford.edu

## Abstract

Euclidean geometry is among the earliest forms of mathematical thinking. While the geometric primitives underlying its constructions, such as perfect lines and circles, do not often occur in the natural world, humans rarely struggle to perceive and reason with them. Will computer vision models trained on natural images show the same sensitivity to Euclidean geometry? Here we explore these questions by studying few-shot generalization in the universe of Euclidean geometry constructions. We introduce *Geoclidean*, a domain-specific language for Euclidean geometry, and use it to generate two datasets of geometric concept learning tasks for benchmarking generalization judgements of humans and machines. We find that humans are indeed sensitive to Euclidean geometry and generalize strongly from a few visual examples of a geometric concept. In contrast, low-level and high-level visual features from standard computer vision models pretrained on natural images do not support correct generalization. Thus Geoclidean represents a novel few-shot generalization benchmark for geometric concept learning, where the performance of humans and of AI models diverge. The Geoclidean framework and dataset are publicly available for download.[*] [†]

## 1 Introduction

The built world we inhabit is constructed from geometric principles. Yet geometric primitives such as perfect lines and circles, which are the foundations of human-made creations, are uncommon in the natural world. Whether for efficiency or for visual aesthetics, whether innate or learned, humans are sensitive to geometric forms and relations. This natural understanding of geometry enables a plethora of applied skills such as design, construction, and visual reasoning; it also scaffolds the development of rigorous mathematical thinking, historically and in modern education. Thus, understanding the visually-grounded geometric universe is an important desideratum for machine vision systems.

Ancient Greek philosophers were amongst the earliest to formalize geometric notions, culminating in Euclid's geometry in the 4th century BC. With a compass and straight edge, Euclid's axioms can construct a geometric world that reflects an idealized, or Platonic, physical reality. We hypothesize that Euclidean constructions are intrinsic to human visual reasoning. We thus build a library to define and render such concepts, allowing systematic exploration of geometric generalization. In this paper, we present Geoclidean, a domain-specific language (DSL) for describing Euclidean primitives and construction rules. Sets of construction rules define concepts that can then be realized into infinitely many rendered images capturing the same abstract geometric model.

---

[*]The Geoclidean framework can be found at https://github.com/joyhsu0504/geoclidean_framework.

[†]Datasets can be found at https://downloads.cs.stanford.edu/viscam/Geoclidean/geoclidean.zip.

36th Conference on Neural Information Processing Systems (NeurIPS 2022) Track on Datasets and Benchmarks.

Figure 1: Rendered realizations of Euclidean geometry concepts from the Geoclidean datasets.

Based on Geoclidean, we introduce two datasets to study few-shot generalization to novel rendered realizations of Euclidean geometry concepts (See Figure 1). To succeed in solving these tasks, one must understand the underlying geometric concept of a set of rendered images. The first dataset, Geoclidean-Elements, covers mathematical definitions from the first book of Euclid's Elements [Simson et al., 1838]. The second dataset, Geoclidean-Constraints, simplifies and more systematically explores possible relationships between primitives. We publicly release both datasets, as well as the dataset generation library based on Geoclidean.

We report findings on the Geoclidean few-shot generalization tasks from human experiments, as well as from evaluation on low-level and high-level visual features from ImageNet-pretrained VGG16 [Deng et al., 2009, Simonyan and Zisserman, 2014], ResNet50 [He et al., 2016], InceptionV3 [Szegedy et al., 2016], and Vision Transformer [Dosovitskiy et al., 2020]. We show that humans significantly outperform pretrained vision models, generalizing in a way that is highly consistent with abstract target concepts. ImageNet-pretrained vision models are not as sensitive to intrinsic geometry and do not as effectively encode these geometric abstractions. Our benchmarking process illustrates this gap between humans and models, establishing Geoclidean as an interesting and challenging generalization task for visual representation and geometric concept learning.

## 2 Foundations of Geoclidean

In this section, we give an overview of the Geoclidean DSL, which builds on foundations of Euclid's axioms for constructing with a compass and straightedge. We present a Python library that renders Euclidean geometry concepts described from our DSL into images. We first describe the Euclidean geometry universe in Section 2.1, and then introduce the Geoclidean language and construction rules in Section 2.2. Finally, we show geometric concept realizations into rendered images in Section 2.3.

### 2.1 Euclidean Geometry Universe

There exist numerous systems of geometry, each with its own logical system. Each logic can be described in a formal language, each formal language describes construction rules, and each set of construction rules describe *concepts* in the geometric universe. Geometric concepts can be realized and rendered into images. Euclidean geometry was among the earliest formalized, first described by Euclid in Elements [Simson et al., 1838]. The first book of Elements details plane geometry, laying the foundation for basic properties and propositions of geometric objects. Importantly, Euclid's geometry is constructive—objects only exist if they can be created with a compass and straightedge. Euclidean constructions are *abstract* models of geometric objects, specified without the use of coordinates and without concrete realizations into images. Hence, changes to the image rendering, such as alterations to size and rotation, do not change an object's intrinsic Euclidean geometry.

Euclid's axioms build the foundations of this geometric universe via 1) draw a straight line from any point to any point, 2) describe a circle with any centre and distance. The Euclidean geometry universe is determined by these base geometric *primitives*, and the *constraints* that parameterize object relationships. There are other universes that contain different logic systems, from Descartes'

analytic geometry, to Euler's affine geometry, to Einstein's special relativity. Because of its simplicity and abstraction we are particularly interested in the universe of Euclidean geometry – do humans and machines find these constructions natural? We later explore whether they spontaneously generalize image classes according to Euclidean rules.

## 2.2 Domain-Specific Language: Geoclidean

We create a domain-specific language (DSL), Geoclidean, defining Euclidean constructions that arise from the mathematics of Euclidean geometry. Geoclidean allows us to define construction rules for objects and their relations to each other, encompassing concepts in the Euclidean geometry universe. It includes three simple primitives. The first is a **point**, parameterized by constraints, if any, to an object or a set of objects previously defined. The point is defined without specific coordinates, and only when *realized*, would be assigned coordinate values $x$ and $y$. The second is a **line**, which, following the first axiom, is parameterized by two points, representing the beginning and end. The third is a **circle** which, following the second axiom, is defined by a center point and an edge point. These primitives represent the compass and straightedge constructions that Euclid introduced. Lines and circles are defined by points, while points can be constrained to previously built objects. In this way primitives are sequentially defined to form a *concept*.

Table 1: The Geoclidean DSL for building Euclidean geometry concepts. We assume a pool of variable names for points and objects (`point_name`, `object_name`). As a shorthand for point creation followed by reference, we later use e.g. `Line(p1(),p2())` to represent `p1 = Point();` `... Line(p1,p2)`. The marker * indicates that the object will not be visible in the final rendering.

| | | |
|---|---|---|
| CONCEPT | $\rightarrow$ | STATEMENT; CONCEPT |
| STATEMENT | $\rightarrow$ | `OBJECT_NAME`VISIBILITY = OBJECT(`POINT_NAME`, `POINT_NAME`) \| |
| | | `POINT_NAME` = POINT(CONSTRAINTS) |
| VISIBILITY | $\rightarrow$ | [] \| * |
| OBJECT | $\rightarrow$ | LINE \| CIRCLE |
| CONSTRAINTS | $\rightarrow$ | [] \| [`OBJECT_NAME`] \| [`OBJECT_NAME`, `OBJECT_NAME`] |

Geoclidean's syntax is defined in Table 1. We can initialize a new point, line, or circle via the constructors `Point`, `Line`, `Circle`, assigning them to a named variable. For points we generally use a shorthand to define and use the variable inline. For example, `p1()` is a free point that is unconstrained to any objects, and can be realized anywhere in the image, `p2(circle1)` is a partially constrained point that lies on the object `circle1`, and `p3(line1, line2)` is a fully constrained point that lives in the intersection of `line1` and `line2`. A point can be reused by referring to its name. The line `Line(p1, p2)` is parameterized by two end points, and the circle `Circle(p1, p2)` is parameterized by points at the center and edge. Not indicated in Table 1 is the semantic constraint that variable names must be defined before being used within constructors. Visibility of rendering is denoted for each object, with * indicating that the object is *not* visible in the final rendering, as some objects in Euclidean geometry are used solely as helper constructions for other core objects in the concept. Each construction rule is a geometric object, and a sequence of construction rules define a Euclidean geometry concept.

## 2.3 Concept Realization

Geoclidean implements this Euclidean language and renders realizations of concepts from the language into images. For each line or circle object, we realize its rendering based on the point parameters required. During rendering, each point parameter is given randomly sampled real-valued coordinates bound by its constraints. If the point exists already, Geoclidean reuses the past component; if it is a free point without constraints, Geoclidean randomly samples values for $x$ and $y$; if constrained, Geoclidean randomly samples a point that lies constrained on the object or the intersection of a set of objects. It is possible for intersection sets to be empty, and we reject sample realizations until finding a satisfying realization for all points. Geoclidean creates the objects sequentially and renders them into an image, if visible.

**Algorithm 1** Construction rules for the equilateral triangle concept, with colored steps corresponding to the rendered realization in Figure 2. The two circles `c1` and `c2` are not rendered in the final image as denoted by *.

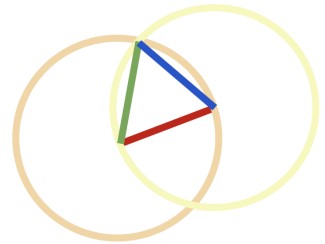

```
1: l1 = Line(p1(), p2())              ▷ Red line
2: c1*= Circle(p1(), p2())           ▷ Light orange circle
3: c2*= Circle(p2(), p1())           ▷ Light yellow circle
4: l2 = Line(p1(), p3(c1, c2))        ▷ Green line
5: l3 = Line(p2(), p3(c1, c2))        ▷ Blue line
```

Figure 2: Rendered realization of the equilateral triangle concept.

We see in Figure 2 an example of the Geoclidean language describing the equilateral triangle concept realized into an image. The construction rules in Algorithm 1 are created step by step; we color each object for clarity. The first construction rule creates a red line between two unconstrained free points, `p1()` and `p2()`; when realized with sampled real-valued points, this line can be anywhere in the image with any length and rotation. The second rule creates an invisible orange circle with the ends of the first line as its center and edge point. The third rule creates another invisible yellow circle with the center and edge point flipped, forming intersecting helper circles with the same radius. Then, the fourth rule creates a green line between p1 and a new point `p3(c1, c2)`, which is a point constrained to the intersection of the previously created orange and yellow circles (the realization randomly chooses one of the two intersection points). The last rule creates the final blue line between p2 and p3, completing the constructed triangle and enforcing all sides to be of equal length. This concept is that of the *equilateral triangle*, and we see that the invisible helper circle objects serve as essential constraints to the final rendering.

Importantly, the construction rules do not specify any coordinates, and our Geoclidean framework creates the coordinates upon realization of the concept into an image. Hence, the rendered equilateral triangle can be of any size and orientation, while always respecting the underlying geometric concept. In Geoclidean, every random realization of this concept creates equilateral triangles, as represented in Euclid's geometric universe. The Geoclidean framework allows us to create rendered image datasets that follow the specified concept language.

## 3   Geoclidean Task and Datasets

Do the realizations of Euclidean constructions form natural categories for humans? For computer vision models? To study these questions we introduce a few-shot generalization task based on Geoclidean and two image datasets that realize 37 Euclidean geometry concepts.

**Task.**   We explore few-shot generalization from positive examples of a target concept; to the extent that participants generalize to other realizations of the target concept, and not realizations of more general concepts, we conclude the concept is an intuitive kind.

For each concept, the task includes five reference examples of the target concept and a test set of 15 images. Among the 15 images, there are five positive examples and two sets of five negative examples. Here, positive examples in the test set are realizations of the target concept, as are the reference examples, and negative examples are realizations of related but different concepts (which are not realizations of the target concept). The goal is to correctly categorize positive examples as positive and negative examples as negative in the test set. The ten negative examples are divided into five *Close* and five *Far* examples, where the negative examples in *Close* are from a closely related concept with a fewer number of constraint differences from the target concept, and negative examples in *Far* are from a further, less related concept with a larger number of constraint differences. These constraint differences consist of altering a point to have fewer constraints compared to that point in the target concept (yielding a more general and less specified geometric concept). See Figure 3 for examples, with the top representing reference examples, and the bottom representing the test set. Note that, because we are interested in intrinsic sensitivity of visual representations to geometric concepts, we are not introducing a meta-learning task: there are no few-shot generalization sets intended for model training.

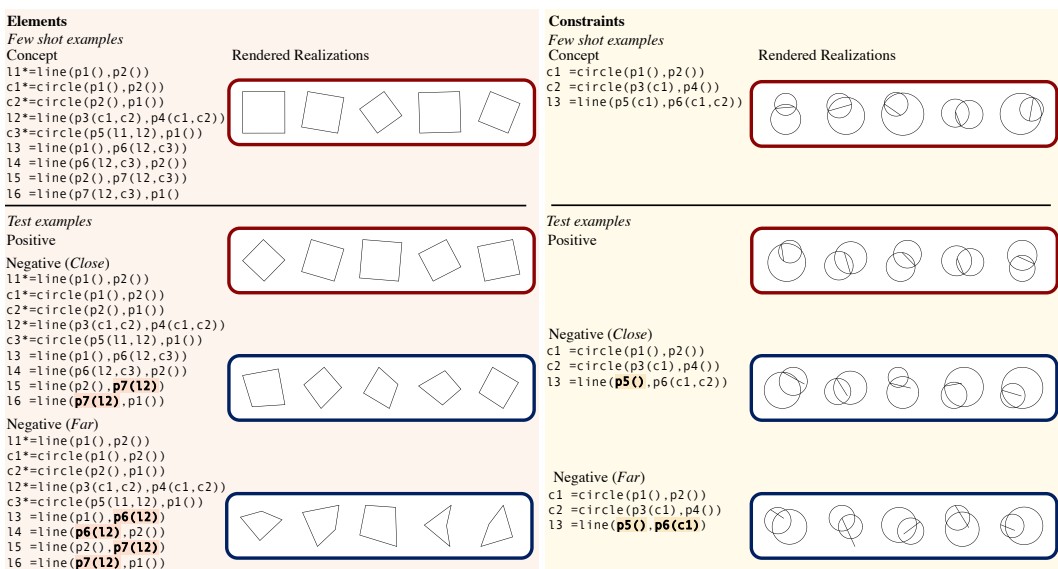

Figure 3: Examples of Geoclidean-Elements and Geoclidean-Constraints tasks. Each task consists of few-shot reference examples as well as test examples. Few-shot examples and positive test examples derive from the same concept (contained in red boxes), while negative test examples derive from a related but different concept (contained in blue boxes); *Close* test examples differ by fewer point differences in the construction rules than *Far*, seen bolded in the Geoclidean language.

We now introduce the two datasets we created based on Geoclidean for the generalization task. The first dataset, Geoclidean-Elements, includes the tests of 17 concepts derived from the first book of Euclid's Elements; the second dataset, Geoclidean-Constraints, includes the tests of 20 concepts based on constraints defining relationships between Geoclidean primitives. See Figure 3 for examples from both splits.

**Geoclidean-Elements.** The Geoclidean-Elements dataset is derived from definitions in the first book of Euclid's Elements, which focuses on plane geometry. Geoclidean-Elements includes 17 target concepts, which, along with Geoclidean primitives, covers definitions in the first book of Elements. These concepts require complex construction rules with helper objects that are not visible in the final renderings. Realizations from Geoclidean-Elements test sensitivity to the exactness of Euclidean constructions without explicit visual constraint differences.

The concepts in Geoclidean-Elements include angle (Book I definition IX), perpendicular bisector (def X), angle bisector (def X), sixty degree angle (def XI and XII), radii (def XV), diameter (def XVII and XVIII), segment (def XIX), rectilinear (def XX and XXIII), triangle (def XXI), quadrilateral (def XXII and XXXIV), equilateral triangle (def XXIV, XXV, XXVI in *Close* and *Far*), right angled triangle (def XXVII, XXVIII, XXIX in *Close* and *Far*), square (def XXX), rhombus (def XXXI), oblong (def XXXII), rhomboid (def XXXIII), and parallel lines (def XXXV). The rest of the definitions are descriptions of Geoclidean primitives (e.g. points, lines).

**Geoclidean-Constraints.** The Geoclidean-Constraints dataset consists of 20 concepts, created from permutations of line and circle construction rules with various constraints describing the relationship between objects. This dataset focuses on explicit constraints between geometric objects. We denote the objects as the following—lines as L, circles as C, and triangles (constructed from three lines) as T. Tasks include three, four, and five object variants, each with specific ordering; the different ordering of objects is significant, as constraints may only depend on previously defined objects. Each concept is defined by object ordering as well as constraints describing the relationships between them; the full set of construction rules for each concept is released with the dataset.

The three object concepts are [LLL, CLL, LLC, CCL, LCC, CCC], the four object concepts are [LLLL, LLLC, CLLL, CLCL, LLCC, CCCL, CLCC, CCCC], and the five object concepts are

`[TLL, LLT, TCL, CLT, TCC, CTT]`. These 20 concepts test the few-shot generalization capability in constrained Euclidean geometry concepts.

## 4    Findings

We present our findings on the Geoclidean dataset in benchmarking human performance (Section 4.1) and pretrained vision models' capabilities (Section 4.2). We show that humans are indeed sensitive to Euclidean geometry concepts, generalizing strongly from five examples across the 37 concepts. This establishes Geoclidean as an interesting task for evaluating the human-like visual competencies of machine vision. Indeed, we find that state-of-the-art pretrained visual representations perform poorly on this few-shot generalization task.

### 4.1    Human Performance

We collected human judgements for the Geoclidean few-shot concept learning task. We recruited 30 participants for each concept using the Prolific crowd-sourcing platform [Palan and Schitter, 2018]. As mentioned above, participants are given five example realizations of the target concept and 15 test images including five positive examples, five *Close* negative examples, and five *Far* negative examples. Each of the test questions states: "Each of the five images shown is a 'wug'. Is the image below a 'wug' or not a 'wug'?", where 'wug' is a random made-up word for each task. The order of tasks and realizations is randomized for each participant to remove order effects. The experiment interface was implemented on Qualtrics, with details described in the Appendix.

We report task accuracy in Table 2, scored as the percentage of participants correctly categorizing test images, averaged across all examples. We split the 15 test images into two tasks, with the *Close* task consisting of 5 positive examples and 5 negative examples from *Close*, and the *Far* task consisting of the same 5 positive images with 5 negative examples from *Far*. Each task contains 10 examples in total. We see that human performance is strong across all tasks, with on-average higher scores in *Far* compared to *Close*, show-

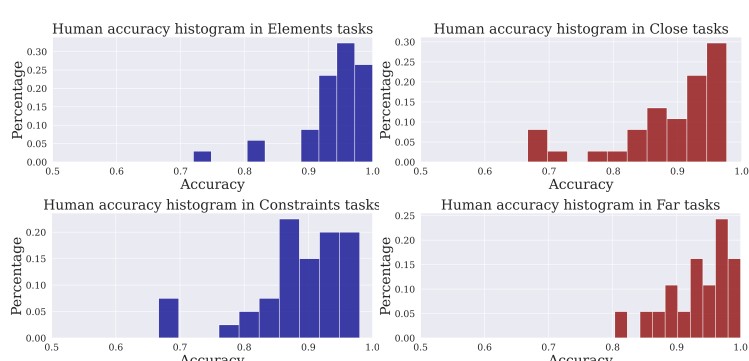

Figure 4: Histogram of human accuracies on the Geoclidean-Elements and the Geoclidean-Constraints datasets, as well as accuracies with *Close* and *Far* negative examples. The y-axis indicates the percentage of tasks with the specified accuracy on the x-axis.

ing that the number of differences in construction rules affects the semantic distance between rendered realizations. Only tasks `LLL, CLCL`, and rhomboid yielded slightly better performance in *Close* than *Far*. Out of all tasks, `LLC, CCC, LLLC` with *Close* negatives are more difficult for humans (with equally poor performance across all test images), which we hypothesize is due to more subtle constraint intersections. In general, humans perform well on this generalization task.

We show accuracy histograms in Figure 4, with the left two plots depicting results from concepts in Geoclidean-Elements and Geoclidean-Constraints, and the right two plots depicting results when calculated with *Close* negative examples and *Far* negative examples. Humans are more sensitive to concepts in Geoclidean-Elements, which are complex constructions that test the exactness of shapes (e.g., squares and equilateral triangles), and slightly less sensitive to concepts in Geoclidean-Constraints, which test the precise relationships between objects (e.g., constrained contact point between the end of a line and the center of a circle).

Participants could generate infinitely many rules consistent with the positive images seen in the few-shot examples (e.g., a "wug" can have its own prototype for each example, as there are no negative examples), and there is potential ambiguity as to which are the correct construction rules

of the concept. Despite this wide range of possible generalization patterns, the generalization rule chosen by humans corresponds well to the Euclidean construction universe.

Table 2: Human accuracy across all 74 tasks in Geoclidean.

| CONCEPT | Close | Far | CONCEPT | Close | Far |
|---|---|---|---|---|---|
| ANGLE | 0.9767 | 0.9833 | LLL | 0.9700 | 0.9667 |
| PERP BISECTOR | 0.9367 | 0.9833 | CLL | 0.9467 | 0.9667 |
| ANG BISECTOR | 0.9433 | 0.9533 | LLC | 0.6767 | 0.9233 |
| SIXTY ANG | 0.8233 | 0.9533 | CCL | 0.8700 | 0.8833 |
| RADII | 0.9233 | 0.9600 | LCC | 0.8867 | 0.9633 |
| DIAMETER | 0.9567 | 1.0000 | CCC | 0.6667 | 0.8767 |
| SEGMENT | 0.9300 | 0.9833 | LLLL | 0.8833 | 0.9767 |
| RECTILINEAR | 0.9000 | 0.9033 | LLLC | 0.6667 | 0.8867 |
| TRIANGLE | 0.9633 | 0.9767 | CLLL | 0.8367 | 0.9033 |
| QUADRILATERAL | 0.9167 | 0.9267 | CLCL | 0.8700 | 0.8567 |
| EQ T | 0.9533 | 0.9800 | LLCC | 0.8867 | 0.9333 |
| RIGHT ANG T | 0.7200 | 0.8133 | CCCL | 0.9233 | 0.9333 |
| SQUARE | 0.8933 | 0.9867 | CLCC | 0.8633 | 0.9000 |
| RHOMBUS | 0.9367 | 0.9667 | CCCC | 0.8167 | 0.8800 |
| OBLONG | 0.9666 | 0.9900 | TLL | 0.9467 | 0.9800 |
| RHOMBOID | 0.9700 | 0.9300 | LLT | 0.9267 | 0.9400 |
| PARALLEL L | 0.9500 | 0.9567 | TCL | 0.9533 | 0.9633 |
| | | | CLT | 0.9533 | 0.9633 |
| | | | TCC | 0.9533 | 0.9633 |
| | | | CCT | 0.9533 | 0.9633 |

## 4.2 Model Benchmarks

We benchmark pretrained vision models' performance on the Geoclidean task, to evaluate few-shot generalization (with no meta-learning or fine-tuning) in the Euclidean geometry universe. We measure performance of features from ImageNet-pretrained VGG16 [Simonyan and Zisserman, 2014], ResNet50 [He et al., 2016], InceptionV3 [Szegedy et al., 2016], and Vision Transformer [Dosovitskiy et al., 2020], and evaluate both low-level features and high-level features for each of the

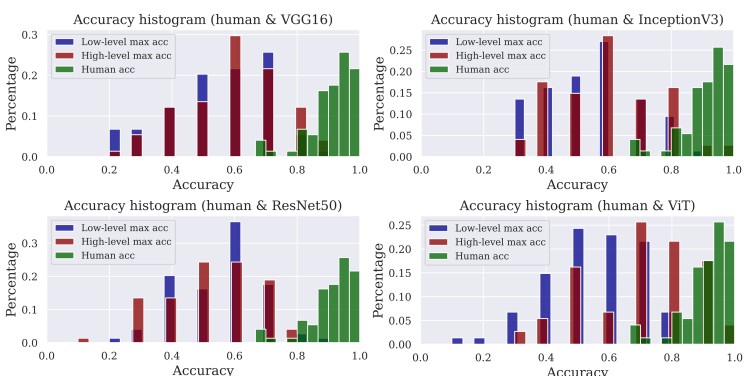

Figure 5: Histogram of human and maximum low-level and high-level feature accuracies of various vision models.

models. Low-level features are outputs of earlier layers in neural networks, which tend to capture low-level information such as edges and primitive shapes, while high-level features are outputs of later layers that tend to capture more high-level semantic information [Zeiler and Fergus, 2014]. We detail how we define the layers for each baseline in the Appendix. To evaluate these features, we extract features $\phi$ from the few-shot $t$ reference images of the target concept, to create a prototype target feature, $T = \frac{1}{n} \sum_{i=1}^{n} \phi(t_i)$. We classify a test image $r$ as in-concept if it is closer than a threshold to the prototype: $f|T - \phi(r)| < \theta$, where $f$ is the normalizing function between $0 \sim 1$. The threshold is fit by selecting the best-performing normalized distance threshold across all 74 tasks, for given features. (By fitting this free threshold, we bias the reported accuracy in favor of models.)

Table 3: Human accuracy compared to low and high-level feature accuracy of vision models across 37 concepts in Geoclidean, each concept containing averaged accuracies between *Close* and *Far* tasks.

| AUC | HUMAN | VGG16 LOW | HIGH | RN50 LOW | HIGH | INV3 LOW | HIGH | VIT LOW | HIGH |
|---|---|---|---|---|---|---|---|---|---|
| ANGLE | 0.98 | 0.50 | 0.45 | 0.45 | 0.50 | 0.40 | 0.45 | 0.50 | 0.40 |
| PERP BISECTOR | 0.96 | 0.70 | 0.85 | 0.70 | 0.50 | 0.65 | 0.90 | 0.70 | 0.85 |
| ANG BISECTOR | 0.95 | 0.50 | 0.60 | 0.50 | 0.65 | 0.60 | 0.60 | 0.50 | 0.75 |
| SIXTY ANG | 0.89 | 0.25 | 0.65 | 0.35 | 0.45 | 0.55 | 0.45 | 0.35 | 0.70 |
| RADII | 0.94 | 0.75 | 0.70 | 0.75 | 0.60 | 0.65 | 0.80 | 0.75 | 0.75 |
| DIAMETER | 0.98 | 0.30 | 0.55 | 0.40 | 0.40 | 0.50 | 0.75 | 0.45 | 0.85 |
| SEGMENT | 0.96 | 0.45 | 0.60 | 0.60 | 0.40 | 0.30 | 0.65 | 0.55 | 0.65 |
| RECTILINEAR | 0.90 | 0.65 | 0.35 | 0.60 | 0.45 | 0.55 | 0.60 | 0.65 | 0.45 |
| TRIANGLE | 0.97 | 0.65 | 0.45 | 0.50 | 0.40 | 0.40 | 0.55 | 0.35 | 0.60 |
| QUADRILATERAL | 0.92 | 0.50 | 0.70 | 0.60 | 0.50 | 0.55 | 0.75 | 0.60 | 0.60 |
| EQ T | 0.97 | 0.40 | 0.85 | 0.50 | 0.70 | 0.65 | 0.65 | 0.50 | 0.55 |
| RIGHT ANG T | 0.77 | 0.65 | 0.75 | 0.60 | 0.70 | 0.60 | 0.55 | 0.70 | 0.55 |
| SQUARE | 0.94 | 0.70 | 0.80 | 0.65 | 0.45 | 0.85 | 0.70 | 0.60 | 0.75 |
| RHOMBUS | 0.95 | 0.70 | 0.55 | 0.60 | 0.55 | 0.60 | 0.55 | 0.70 | 0.60 |
| OBLONG | 0.98 | 0.45 | 0.55 | 0.45 | 0.45 | 0.50 | 0.70 | 0.45 | 0.70 |
| RHOMBOID | 0.95 | 0.70 | 0.60 | 0.65 | 0.55 | 0.50 | 0.50 | 0.65 | 0.75 |
| PARALLEL L | 0.95 | 0.55 | 0.35 | 0.50 | 0.40 | 0.45 | 0.55 | 0.45 | 0.85 |
| LLL | 0.97 | 0.50 | 0.70 | 0.55 | 0.40 | 0.55 | 0.35 | 0.55 | 0.80 |
| CLL | 0.96 | 0.25 | 0.60 | 0.40 | 0.30 | 0.30 | 0.60 | 0.30 | 0.60 |
| LLC | 0.80 | 0.65 | 0.55 | 0.60 | 0.55 | 0.55 | 0.65 | 0.60 | 0.60 |
| CCL | 0.88 | 0.65 | 0.50 | 0.55 | 0.65 | 0.30 | 0.60 | 0.55 | 0.75 |
| LCC | 0.93 | 0.45 | 0.70 | 0.45 | 0.70 | 0.65 | 0.70 | 0.45 | 0.85 |
| CCC | 0.77 | 0.60 | 0.55 | 0.60 | 0.70 | 0.70 | 0.75 | 0.60 | 0.90 |
| LLLL | 0.93 | 0.60 | 0.30 | 0.50 | 0.30 | 0.30 | 0.50 | 0.50 | 0.50 |
| LLLC | 0.78 | 0.30 | 0.65 | 0.40 | 0.55 | 0.50 | 0.55 | 0.30 | 0.80 |
| CLLL | 0.87 | 0.55 | 0.65 | 0.60 | 0.65 | 0.65 | 0.50 | 0.70 | 0.65 |
| CLCL | 0.86 | 0.60 | 0.65 | 0.65 | 0.45 | 0.40 | 0.35 | 0.60 | 0.85 |
| LLCC | 0.91 | 0.75 | 0.65 | 0.85 | 0.55 | 0.80 | 0.70 | 0.80 | 0.70 |
| CCCL | 0.93 | 0.35 | 0.60 | 0.50 | 0.65 | 0.35 | 0.65 | 0.45 | 0.75 |
| CLCC | 0.88 | 0.60 | 0.55 | 0.55 | 0.60 | 0.60 | 0.75 | 0.55 | 0.80 |
| CCCC | 0.85 | 0.80 | 0.65 | 0.70 | 0.60 | 0.75 | 0.50 | 0.75 | 0.95 |
| TLL | 0.96 | 0.70 | 0.65 | 0.65 | 0.60 | 0.60 | 0.55 | 0.75 | 0.85 |
| LLT | 0.93 | 0.55 | 0.40 | 0.55 | 0.65 | 0.55 | 0.65 | 0.50 | 0.85 |
| TCL | 0.96 | 0.65 | 0.70 | 0.70 | 0.55 | 0.60 | 0.60 | 0.65 | 0.45 |
| CLT | 0.88 | 0.50 | 0.60 | 0.45 | 0.50 | 0.55 | 0.65 | 0.45 | 0.70 |
| TCC | 0.84 | 0.30 | 0.45 | 0.35 | 0.35 | 0.55 | 0.45 | 0.30 | 0.45 |
| CCT | 0.79 | 0.55 | 0.70 | 0.45 | 0.60 | 0.60 | 0.55 | 0.50 | 0.85 |
| **AVERAGE** | **0.91** | **0.55** | **0.60** | **0.55** | **0.53** | **0.54** | **0.60** | **0.55** | **0.70** |

In Table 3, we present accuracy of ImageNet-pretrained low-level and high-level features across different models. Humans substantially outperform features from vision models, showcasing the gap between human and model capabilities in Euclidean geometry concept learning.

We see that, on average across tasks, pretrained vision models perform poorly compared to humans. In general, high-level features perform slightly better than low-level features, though there are cases where this differs. Interestingly, the high-level features from the Vision Transformer (ViT) outperform its convolutional network counterparts, and achieve human-level performance in some tasks. There are a few tasks where the visual feature accuracy outperforms humans, notably tasks CCC, LLLC, CCCC, CCT, where high-level features from ViT perform strongly. We hypothesize that this is because humans are not as sensitive to details of intersections between circles.

In Figure 5, we present histograms comparing maximum low and high-level feature accuracy to human accuracy, illustrating the gap in performance. In Table 4, we report the Pearson correlation coefficient between the answers of humans and models. We see that humans are generally more aligned with high-level features than low-level ones. Additionally, though ViT achieves high accuracy,

it is not correlated to human performance, indicating failure to generalize in the most human-like way.

Table 4: Pearson's correlation to human accuracy across all 74 tasks in Geoclidean.

| VGG16 LOW | HIGH | RN50 LOW | HIGH | INV3 LOW | HIGH | VIT LOW | HIGH |
|---|---|---|---|---|---|---|---|
| -0.08947 | 0.0943 | 0.0341 | -0.1522 | -0.0820 | 0.2575 | -0.0056 | -0.0259 |

We report additional comparisons of low-level and high-level visual features from ImageNet-pretrained VGG16, ResNet50, InceptionV3, and Vision Transformer in the Appendix, and show that similar trends follow across data splits and performance metrics. We also include low-level and high-level feature visualizations in the Appendix, comparing Geoclidean tasks that require reasoning, to perception tasks involving simple geometric primitives and perturbations. These comparisons highlight Geoclidean as a unique and interesting test for vision models.

## 5 Related Work

**Geometric reasoning datasets.** Prior geometric datasets generally fall into two main categories—with geometric objects for computer-aided design (CAD) and for plane geometry. In the first category, the SketchGraphs dataset models relational geometry in CAD design [Seff et al., 2020], and the ABC-Dataset includes parametric representations of 3D CAD models [Koch et al., 2019]. In the latter category, CSGNet presented a generated dataset of constructive solid geometry based on 2D and 3D synthetic programs with squares, circles, and triangles [Sharma et al., 2018], while Ellis et al. [2018] connected high–level Latex graphics programs with 2D geometric drawings. Works such as Zhang et al. [2022], Lu et al. [2021] proposed using datasets with annotated geometric primitives and relationships such as containment from geometry diagrams in textbooks. Others introduced reasoning benchmarks with geometric shapes, including Raven's progressive matrices [Matzen et al., 2010, Wang and Su, 2015, Barrett et al., 2018, Zhang et al., 2019], Bongard problems [Depeweg et al., 2018, Nie et al., 2020], odd-one-out tasks [Mańdziuk and Żychowski, 2019], and a variety of reasoning challenges [Hill et al., 2019, Zhao et al., 2021, El Korchi and Ghanou, 2020, Zhang et al., 2020]. Our work is more related to the latter of geometric shapes, and Geoclidean differs by targeting Euclidean geometry concept learning whose construction language 1) does not require specific coordinates, and 2) focuses on the construction steps that form semantically-complex geometric concepts and the constraints between geometric primitives that humans are intrinsically sensitive to.

**Few-shot concept learning.** Few-shot learning tasks range in complexity on both the input and task description axis. In the natural language processing domain, tasks such as FewRel [Han et al., 2018] and Few-NERD [Ding et al., 2021] have been proposed for few-shot relation classification and entity recognition. Goodman et al. [2008] introduced concept learning tasks with sequences generated from specified logical rules. In the vision domain, which we are interested in, commonly used tasks include those from Lake et al. [2015], which introduced Omniglot as a collection of simple visual concepts collected from 50 writing systems, and from Vinyals et al. [2016], which proposed miniImageNet [Deng et al., 2009], both for the task of one-shot classification. Works in other vision domains include Massiceti et al. [2021], which explores the few-shot video recognition challenge, and Xiao et al. [2020], Gehler et al. [2008], which examines the few-shot color constancy problem. Triantafillou et al. [2019] created the meta-dataset as a diverse dataset for few-shot learning, with multiple tasks for meta-training such as Maji et al. [2013], Wah et al. [2011], Cimpoi et al. [2014], Nilsback and Zisserman [2008], Houben et al. [2013], Lin et al. [2014]. In comparison, we propose Geoclidean as a zero-shot meta-trained, few-shot generalization task that consists of labeled image renderings from a single target concept.

## 6 Discussion

An important contribution of our task is that it allows for better testing of vision models that aim to incorporate reasoning and high-level semantics. Additionally, Geoclidean's zero-shot meta-trained evaluation is especially significant, as many downstream tasks that may leverage pretrained models would greatly benefit from geometric reasoning, such as construction (LegoTron [Walsman et al.,

2022], Physical Construction Tasks Bapst et al. [2019]), physical reasoning (CLEVRER [Yi et al., 2019], ThreeDWorld [Gan et al., 2020]), and shape understanding tasks (PartNet [Mo et al., 2019], ShapeNet [Chang et al., 2015]). Furthermore, numerous additional evaluation tasks can be built with the Geoclidean DSL and rendering library, such as those involving natural language or generated large-scale datasets. We include further analyses and discussion in the Appendix.

# 7 Conclusion

We have introduced Geoclidean, a domain-specific language for the realization of the Euclidean geometry universe, and presented two datasets of few-shot concept learning to test generalization capability in the geometry domain. Humans considerably outperform vision models on Geoclidean tasks, and we believe that this gap illustrates the potential for improvement in learning visual features that align with human sensitivities to geometry. Geoclidean is thus an important generalization task that vision models are not yet sensitive to, and an effective benchmark for geometric concept learning. Furthermore, such explorations of geometric generalization may help us to understand how human vision made the leap from natural forms to the Platonic forms so prevalent in modern design and engineering.

We expect minimal negative societal impact from the release of Geoclidean. We hope future work can build on the foundations of Geoclidean for augmenting vision models in areas such as geometric reasoning and construction, as well as in applications such as education, where geometry is both an essential academic subject and an introduction to proof-based mathematics.

**Acknowledgements** We thank Gabriel Poesia and Stephen Tian for providing valuable feedback on the paper. This work is in part supported by the Stanford Institute for Human-Centered Artificial Intelligence (HAI), Center for Integrated Facility Engineering (CIFE), Analog, Autodesk, IBM, JPMC, Salesforce, and Samsung. JH is supported by the Knight Hennessy fellowship and the NSF Graduate Research Fellowship.

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
