# Supplementary Material for Geoclidean: Few-Shot Generalization in Euclidean Geometry

## A Appendix

We provide access to the Geoclidean framework in Section A.1, data download and usage instructions in Section A.2, and details on human experiments in Section A.3. Analyses on feature visualizations can be found in Section A.4, and analyses on pretraining in Section A.5. We include additional model benchmarking results in Section A.6, discussion on potential use cases in Section A.7, and the full Geoclidean datasheet in Section A.8.

### A.1 Framework

The Geoclidean framework is released at https://github.com/joyhsu0504/geoclidean_framework/, we include tutorials on how to create new image renderings from the Geoclidean domain-specific language.

### A.2 Data

Our dataset is licensed under CC-BY 4.0. Construction rules and rendered images for all concepts in the Geoclidean-Elements and Geoclidean-Constraints can be found at https://downloads.cs.stanford.edu/viscam/Geoclidean/geoclidean.zip. The image dataset is easy to download and simple to use; in addition, https://github.com/joyhsu0504/geoclidean_framework/ includes details on the structured data and tutorials on how the dataset can be read and processed. Users can create additional image datasets with the released framework should they wish to explore more Euclidean geometry concepts.

The Geoclidean dataset is hosted on the Stanford CS cluster for long term preservation, publicly accessible by all and with no plans for future changes. We bear all responsibility in case of violation of rights. The full datasheet for Geoclidean can be found in Section A.8. The dataset also lives at https://doi.org/10.5281/zenodo.6643898.

### A.3 Human Experiments

Participants were recruited on Prolific [Palan and Schitter, 2018], and compensated with an hourly wage of $15.8. The total amount spent on participant compensation was approximately $420. See Figure 1 for an example of a survey question given to participants, hosted on Qualtrics.

Participants voluntarily signed up on Prolific; no personally identifiable information were taken and no offensive content were shown. Participants were given the following IRB notification: "By answering the following questions, you are participating in a study being performed by cognitive scientists in the Stanford Department

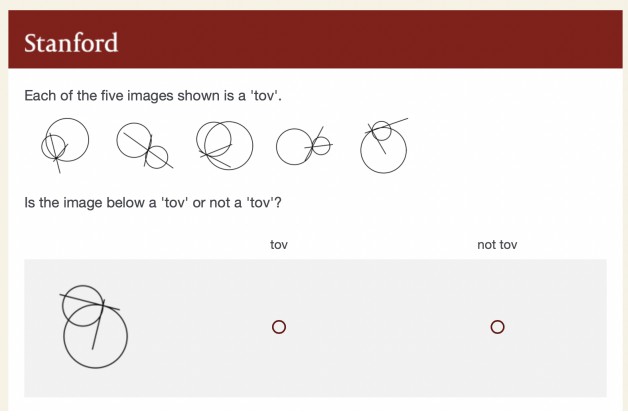

Figure 1: Example of survey question given to participants.

of Psychology. If you have questions about this research, please contact Joy Hsu at joycj@stanford.edu or Noah Goodman at ngoodman@stanford.edu. You must be at least 18 years old to participate. Your participation in this research is voluntary. You may decline to answer any or all of the following questions. You may decline further participation, at any time, without adverse consequences. Your anonymity is assured; the researchers who have requested your participation will not receive any personal information about you".

## A.4    Feature Visualizations

We provide visualizations of low-level features from ResNet50 and the Vision Transformer on a variety of rendered Geoclidean images. In Figure 2, we investigate three comparisons of feature representation space, with PCA dimension reduction applied to each feature to plot in two-dimensional space. The three comparisons are the following: 1) on circles and lines, the most simple geometric primitives, 2) on two different complex Geoclidean concepts which contain different primitives, two circles and one line vs. two lines and one circle, and 3) on positive and negative examples from a Geoclidean task, which contains renderings from related concepts that differ by a few constraint differences with the same primitives.

We see that for both ResNet50 and ViT low-level features, the first two tasks involving different geometric structures achieve strong separation; this holds true for both the simpler and complex case, showing that low-level features do generalize with respect to some visual geometric differences. In contrast, the third Geoclidean task is difficult for both vision models, highlighting the intended difficulty of Euclidean geometric reasoning – why our task is especially interesting. We note that low-level features achieve the same accuracy for both ResNet50 and Vision Transformers, with $0.55$ accuracy, as seen in the main text.

We also provide additional analysis on near-far categorization of the high-level feature space. We apply PCA dimension reduction on high-level features from ResNet50 and study comparisons across different rendered images, for concepts in both Elements and Constraints dataset. In Figure 3, we see feature visualizations from six different groups of rendered images for each base concept. The red, orange, green, blue groups are derived from the Geoclidean task, which include positive examples of the target concept (red and orange), as well as negative examples from related concepts (Far as green and Close as blue). The purple and pink groups are simple perturbations of the target train images: scaled (purple) and warped (pink).

As seen in Figure 3, ResNet50 features are robust to simple perturbations in geometry, as purple and pink images form distinct clusters for near-far categorization. In contrast, the rendered images from related concepts are difficult to distinguish from the target concept, as red, orange, green, and blue are not disentangle-able. While these vision models do learn near and far categorizations well from the geometric perspective, geometric reasoning in Geoclidean is significantly more difficult.

## A.5    Pretraining

We include Geoclidean evaluation of vision models pretrained on the Google Quick Draw dataset [Ha and Eck, 2017, goo], which contains images with line drawings. We trained the VGG16, ResNet50, InceptionV3, and Vision Transformer model on the Google Quick Draw dataset, and these models achieved a top-1 candidate test accuracy of $0.5455$, $0.6752$, $0.6921$, and $0.7039$, respectively. These values are in the same range as the reported value of approximately $0.7$ for the released recurrent neural network in the official Github repository.

In Table 1, we report the accuracy for Quick Draw pretrained models across all Geoclidean tasks, for each vision model and for low-level and high-level features. Compared to results from the main text, we see that ImageNet pretrained models are comparable with Quick Draw–pretrained models in low-level features, and outperform Quick Draw–pretrained models in high-level features. We hypothesize that though the visual domain shift from Quick Draw to Geoclidean is smaller than that of ImageNet to Geoclidean, there is added complexity of geometric reasoning in our proposed tasks that is not captured by drawing classification.

We additionally include evaluation of vision models with randomly initialized parameters. In Table 2, we report the accuracy for randomly initialized vision models across all Geoclidean tasks. High-level features from ImageNet pretrained models consistently outperform features from the randomly

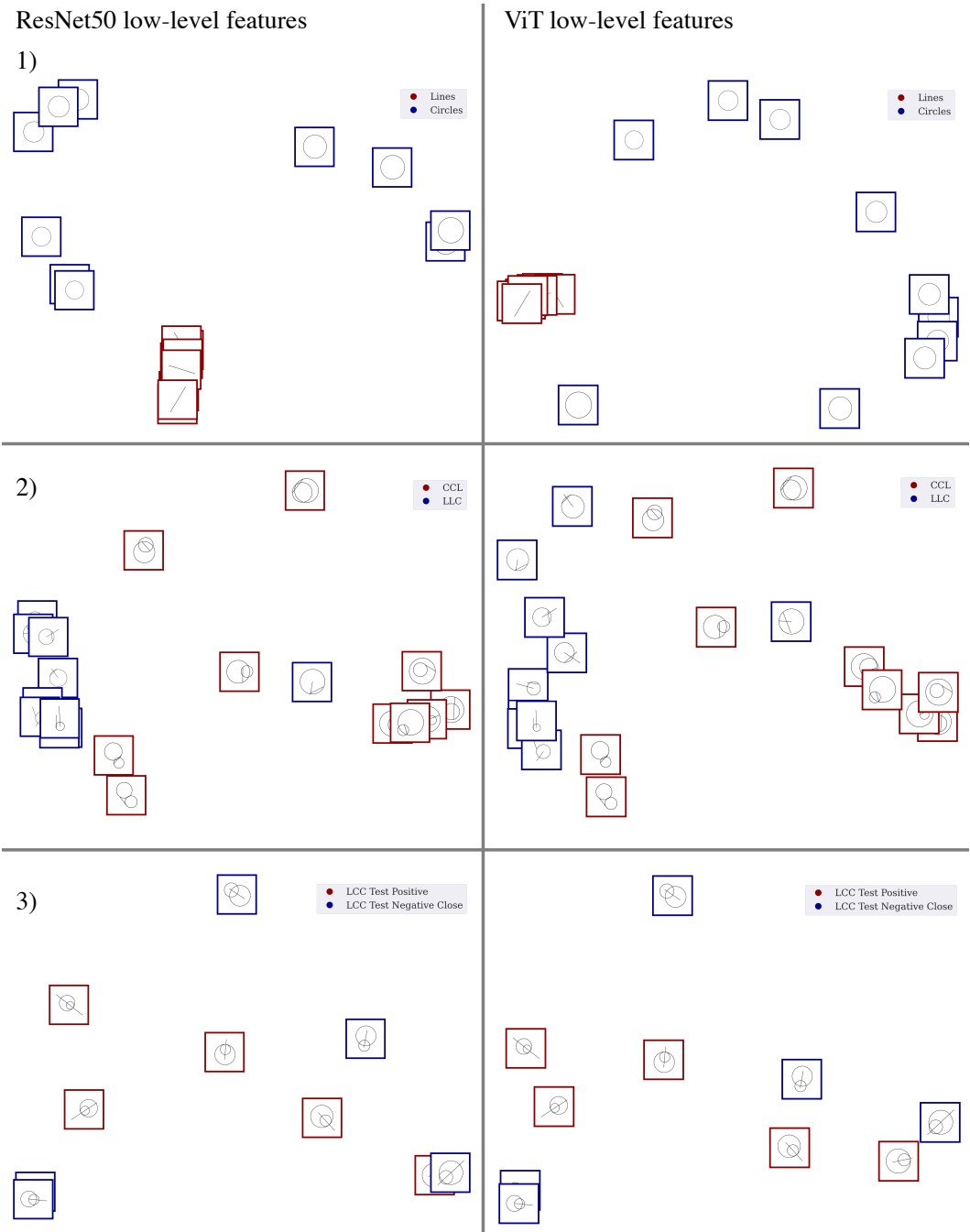

Figure 2: Low-level feature visualizations for ResNet50 and ViT, across a variety of Geoclidean rendered images.

initialized models. We conjecture that this is due to the higher-level semantic information encoded in later layers that capture some capability to conduct geometric reasoning.

## A.6 Model Benchmarks

Code for benchmarking experiments on low-level and high-level features from VGG16 [Simonyan and Zisserman, 2014], ResNet50 [He et al., 2016], InceptionV3 [Szegedy et al., 2016], and Vision Transformer [Dosovitskiy et al., 2020] can be found at https://github.com/joyhsu0504/geoclidean_framework/.

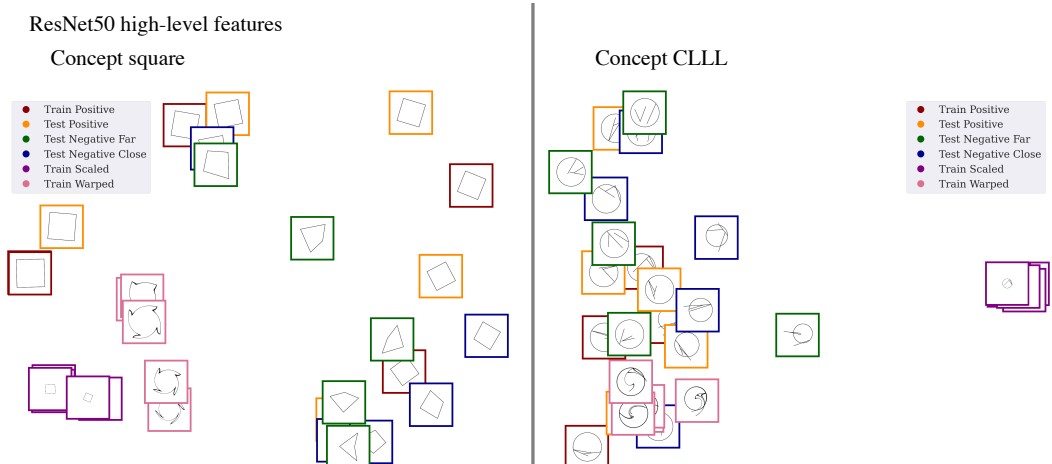

Figure 3: High-level feature visualizations for ResNet50, for rendered images from Geoclidean tasks and from simple geometric perturbations.

We used ImageNet-pretrained convolutional models from the Keras library [Deng et al., 2009], and Vision Transformer from HuggingFace. Only inference was conducted and results are reproducible via the above url. We used 1 TitanRTX GPU for inference. For ImageNet-pretrained VGG16 network, we extract low-level features from the 6th layer and high-level features from the 18th layer. For ResNet50, we use layers 15th and 168th, and for InceptionV3 we use layers 31th and 284th, chosen to be outputs of convolutional layers in earlier blocks and later blocks of the network. For Vision Transformer, we take output from the first ViT encoder and the last logits layer.

We report additional results comparing human and vision model performance on data splits of *Close* and *Far* across Geoclidean-Elements and Geoclidean-Constraints. We use both accuracy calculated as described in the main text as well as AUC as performance metrics. To calculate AUC for vision models, we similarly extract features $\phi$ from the few-shot $t$ reference images of the target concept, to create a prototype target feature, $T = \frac{1}{n} \sum_{i=1}^{n} \phi(t_i)$. We use the distance $|T - \phi(r)|$ for each test image $r$ as thresholds in AUC calculation. For human performance, we use an ensemble of humans as a proxy for human confidence. We take the percentage of all participants that answer positively for a given test image to rank test item distance to the target concept, and use this participant positive response rate as threshold for AUC calculation. See results of accuracy on *Close* tasks in Table 3, accuracy on *Far* tasks in Table 4, AUC on *Close* tasks in Table 5, and AUC on *Far* tasks in Table 6. For all tables, average human performance considerably outperforms that of all models.

## A.7 Discussion

An important contribution of our task is that it will allow for better testing of vision models that aim to incorporate reasoning and high-level semantics. Additionally, Geoclidean's zero-shot meta-trained evaluation is especially significant, as many downstream tasks that may leverage pretrained models would greatly benefit from geometric reasoning, such as construction (LegoTron [Walsman et al., 2022], Physical Construction Tasks Bapst et al. [2019]), physical reasoning (CLEVRER [Yi et al., 2019], ThreeDWorld [Gan et al., 2020]), and shape understanding tasks (PartNet [Mo et al., 2019], ShapeNet [Chang et al., 2015]).

Furthermore, numerous additional evaluation tasks can be built on the Geoclidean framework, such as those involving language models applied to the Geoclidean DSL, or those involving training on a generated large-scale dataset. Follow-up research can leverage Geoclidean to generate instances that cover geometrically equivalent concepts. Additional research directions, such as disentangled representation learning, can also benefit from Geoclidean's ability to flexibly build geometric objects with different interpolations across factors of variation [Locatello et al., 2019, Bengio et al., 2013].

Table 1: Human accuracy compared to low and high-level feature accuracy of vision models with pretrained Quick Draw weights, across 34 concepts in Geoclidean, each concept containing averaged accuracies between *Close* and *Far* tasks.

| AUC | HUMAN | VGG16 LOW | VGG16 HIGH | RN50 LOW | RN50 HIGH | INV3 LOW | INV3 HIGH | VIT LOW | VIT HIGH |
|---|---|---|---|---|---|---|---|---|---|
| ANGLE | 0.98 | 0.55 | 0.45 | 0.50 | 0.50 | 0.40 | 0.60 | 0.50 | 0.60 |
| PERP BISECTOR | 0.96 | 0.70 | 0.75 | 0.65 | 0.50 | 0.65 | 0.70 | 0.70 | 0.65 |
| ANG BISECTOR | 0.95 | 0.55 | 0.50 | 0.40 | 0.45 | 0.50 | 0.45 | 0.50 | 0.60 |
| SIXTY ANG | 0.89 | 0.40 | 0.35 | 0.55 | 0.50 | 0.45 | 0.50 | 0.35 | 0.60 |
| RADII | 0.94 | 0.75 | 0.70 | 0.65 | 0.65 | 0.65 | 0.65 | 0.75 | 0.50 |
| DIAMETER | 0.98 | 0.30 | 0.30 | 0.35 | 0.30 | 0.50 | 0.20 | 0.30 | 0.60 |
| SEGMENT | 0.96 | 0.35 | 0.55 | 0.35 | 0.50 | 0.55 | 0.60 | 0.55 | 0.55 |
| RECTILINEAR | 0.90 | 0.70 | 0.65 | 0.55 | 0.60 | 0.60 | 0.75 | 0.65 | 0.35 |
| TRIANGLE | 0.97 | 0.55 | 0.50 | 0.50 | 0.55 | 0.40 | 0.40 | 0.40 | 0.60 |
| QUADRILATERAL | 0.92 | 0.65 | 0.45 | 0.60 | 0.65 | 0.70 | 0.45 | 0.55 | 0.40 |
| EQ T | 0.97 | 0.20 | 0.40 | 0.55 | 0.45 | 0.55 | 0.60 | 0.50 | 0.65 |
| RIGHT ANG T | 0.77 | 0.60 | 0.35 | 0.65 | 0.45 | 0.45 | 0.70 | 0.70 | 0.60 |
| SQUARE | 0.94 | 0.55 | 0.80 | 0.70 | 0.65 | 0.65 | 0.80 | 0.65 | 0.45 |
| RHOMBUS | 0.95 | 0.60 | 0.55 | 0.50 | 0.60 | 0.70 | 0.40 | 0.70 | 0.55 |
| OBLONG | 0.98 | 0.55 | 0.50 | 0.50 | 0.60 | 0.60 | 0.35 | 0.45 | 0.45 |
| RHOMBOID | 0.95 | 0.45 | 0.50 | 0.50 | 0.55 | 0.50 | 0.65 | 0.65 | 0.60 |
| PARALLEL L | 0.95 | 0.55 | 0.60 | 0.45 | 0.60 | 0.65 | 0.50 | 0.50 | 0.60 |
| LLL | 0.97 | 0.45 | 0.45 | 0.50 | 0.45 | 0.50 | 0.55 | 0.55 | 0.55 |
| CLL | 0.96 | 0.20 | 0.30 | 0.40 | 0.30 | 0.40 | 0.50 | 0.25 | 0.70 |
| LLC | 0.80 | 0.55 | 0.75 | 0.50 | 0.50 | 0.45 | 0.40 | 0.55 | 0.65 |
| CCL | 0.88 | 0.60 | 0.60 | 0.50 | 0.70 | 0.45 | 0.50 | 0.55 | 0.40 |
| LCC | 0.93 | 0.40 | 0.75 | 0.60 | 0.65 | 0.70 | 0.75 | 0.40 | 0.50 |
| CCC | 0.77 | 0.60 | 0.60 | 0.65 | 0.60 | 0.70 | 0.45 | 0.60 | 0.60 |
| LLLL | 0.93 | 0.70 | 0.50 | 0.60 | 0.45 | 0.45 | 0.35 | 0.50 | 0.70 |
| LLLC | 0.78 | 0.45 | 0.45 | 0.50 | 0.55 | 0.60 | 0.65 | 0.30 | 0.55 |
| CLLL | 0.87 | 0.55 | 0.75 | 0.65 | 0.60 | 0.60 | 0.60 | 0.70 | 0.60 |
| CLCL | 0.86 | 0.55 | 0.60 | 0.40 | 0.55 | 0.50 | 0.40 | 0.60 | 0.80 |
| LLCC | 0.91 | 0.80 | 0.45 | 0.60 | 0.40 | 0.70 | 0.70 | 0.80 | 0.55 |
| CCCL | 0.93 | 0.35 | 0.35 | 0.65 | 0.55 | 0.65 | 0.40 | 0.55 | 0.60 |
| CLCC | 0.88 | 0.40 | 0.60 | 0.55 | 0.55 | 0.60 | 0.65 | 0.55 | 0.65 |
| CCCC | 0.85 | 0.60 | 0.85 | 0.70 | 0.55 | 0.65 | 0.80 | 0.80 | 0.50 |
| TLL | 0.96 | 0.65 | 0.60 | 0.50 | 0.65 | 0.65 | 0.75 | 0.70 | 0.35 |
| LLT | 0.93 | 0.60 | 0.45 | 0.60 | 0.45 | 0.60 | 0.70 | 0.55 | 0.70 |
| TCL | 0.96 | 0.55 | 0.45 | 0.65 | 0.60 | 0.55 | 0.65 | 0.70 | 0.65 |
| CLT | 0.88 | 0.40 | 0.60 | 0.50 | 0.50 | 0.55 | 0.20 | 0.45 | 0.60 |
| TCC | 0.84 | 0.45 | 0.40 | 0.60 | 0.50 | 0.55 | 0.45 | 0.30 | 0.65 |
| CCT | 0.79 | 0.55 | 0.65 | 0.65 | 0.55 | 0.55 | 0.45 | 0.50 | 0.50 |
|  | 0.91 | 0.52 | 0.54 | 0.55 | 0.53 | 0.56 | 0.55 | 0.55 | 0.57 |
| **AVERAGE** | **0.91** | **0.52** | **0.54** | **0.55** | **0.53** | **0.56** | **0.55** | **0.55** | **0.57** |

Table 2: Human accuracy compared to low and high-level feature accuracy of vision models with initialized weights across 34 concepts in Geoclidean, each concept containing averaged accuracies between *Close* and *Far* tasks.

| AUC | HUMAN | VGG16 | | RN50 | | INV3 | | VIT | |
|---|---|---|---|---|---|---|---|---|---|
| | | LOW | HIGH | LOW | HIGH | LOW | HIGH | LOW | HIGH |
| ANGLE | 0.98 | 0.60 | 0.35 | 0.50 | 0.35 | 0.45 | 0.55 | 0.60 | 0.50 |
| PERP BISECTOR | 0.96 | 0.60 | 0.70 | 0.70 | 0.70 | 0.80 | 0.85 | 0.70 | 0.45 |
| ANG BISECTOR | 0.95 | 0.45 | 0.45 | 0.50 | 0.35 | 0.50 | 0.65 | 0.40 | 0.60 |
| SIXTY ANG | 0.89 | 0.40 | 0.40 | 0.30 | 0.15 | 0.65 | 0.65 | 0.40 | 0.65 |
| RADII | 0.94 | 0.75 | 0.65 | 0.75 | 0.65 | 0.70 | 0.50 | 0.75 | 0.60 |
| DIAMETER | 0.98 | 0.35 | 0.40 | 0.40 | 0.40 | 0.45 | 0.45 | 0.40 | 0.60 |
| SEGMENT | 0.96 | 0.40 | 0.55 | 0.40 | 0.55 | 0.50 | 0.55 | 0.55 | 0.60 |
| RECTILINEAR | 0.90 | 0.60 | 0.60 | 0.60 | 0.50 | 0.45 | 0.45 | 0.60 | 0.65 |
| TRIANGLE | 0.97 | 0.50 | 0.30 | 0.55 | 0.60 | 0.60 | 0.45 | 0.40 | 0.50 |
| QUADRILATERAL | 0.92 | 0.50 | 0.40 | 0.55 | 0.55 | 0.40 | 0.65 | 0.55 | 0.55 |
| EQ T | 0.97 | 0.40 | 0.60 | 0.50 | 0.60 | 0.75 | 0.85 | 0.40 | 0.40 |
| RIGHT ANG T | 0.77 | 0.60 | 0.75 | 0.70 | 0.70 | 0.75 | 0.60 | 0.60 | 0.70 |
| SQUARE | 0.94 | 0.60 | 0.70 | 0.80 | 0.80 | 0.70 | 0.75 | 0.60 | 0.40 |
| RHOMBUS | 0.95 | 0.60 | 0.75 | 0.60 | 0.40 | 0.60 | 0.65 | 0.60 | 0.35 |
| OBLONG | 0.98 | 0.50 | 0.45 | 0.45 | 0.50 | 0.45 | 0.45 | 0.55 | 0.45 |
| RHOMBOID | 0.95 | 0.50 | 0.50 | 0.60 | 0.60 | 0.70 | 0.55 | 0.50 | 0.70 |
| PARALLEL L | 0.95 | 0.55 | 0.50 | 0.50 | 0.55 | 0.40 | 0.45 | 0.50 | 0.35 |
| LLL | 0.97 | 0.50 | 0.55 | 0.50 | 0.50 | 0.55 | 0.50 | 0.50 | 0.65 |
| CLL | 0.96 | 0.25 | 0.45 | 0.40 | 0.30 | 0.20 | 0.25 | 0.20 | 0.45 |
| LLC | 0.80 | 0.60 | 0.50 | 0.65 | 0.50 | 0.45 | 0.45 | 0.55 | 0.60 |
| CCL | 0.88 | 0.55 | 0.40 | 0.55 | 0.50 | 0.45 | 0.45 | 0.50 | 0.70 |
| LCC | 0.93 | 0.35 | 0.70 | 0.40 | 0.70 | 0.50 | 0.55 | 0.40 | 0.35 |
| CCC | 0.77 | 0.60 | 0.65 | 0.60 | 0.65 | 0.70 | 0.60 | 0.60 | 0.70 |
| LLLL | 0.93 | 0.55 | 0.40 | 0.50 | 0.45 | 0.50 | 0.40 | 0.55 | 0.55 |
| LLLC | 0.78 | 0.40 | 0.65 | 0.45 | 0.40 | 0.55 | 0.75 | 0.40 | 0.45 |
| CLLL | 0.87 | 0.60 | 0.65 | 0.60 | 0.65 | 0.65 | 0.55 | 0.60 | 0.50 |
| CLCL | 0.86 | 0.50 | 0.55 | 0.60 | 0.40 | 0.45 | 0.45 | 0.50 | 0.30 |
| LLCC | 0.91 | 0.75 | 0.65 | 0.75 | 0.75 | 0.65 | 0.50 | 0.65 | 0.55 |
| CCCL | 0.93 | 0.40 | 0.60 | 0.40 | 0.30 | 0.55 | 0.55 | 0.50 | 0.50 |
| CLCC | 0.88 | 0.50 | 0.60 | 0.55 | 0.45 | 0.65 | 0.60 | 0.45 | 0.50 |
| CCCC | 0.85 | 0.60 | 0.70 | 0.75 | 0.75 | 0.70 | 0.45 | 0.60 | 0.55 |
| TLL | 0.96 | 0.70 | 0.65 | 0.65 | 0.60 | 0.75 | 0.50 | 0.70 | 0.45 |
| LLT | 0.93 | 0.60 | 0.50 | 0.60 | 0.65 | 0.60 | 0.40 | 0.50 | 0.55 |
| TCL | 0.96 | 0.70 | 0.60 | 0.70 | 0.65 | 0.65 | 0.55 | 0.65 | 0.45 |
| CLT | 0.88 | 0.50 | 0.60 | 0.45 | 0.65 | 0.60 | 0.65 | 0.45 | 0.55 |
| TCC | 0.84 | 0.45 | 0.50 | 0.30 | 0.35 | 0.55 | 0.55 | 0.35 | 0.75 |
| CCT | 0.79 | 0.50 | 0.50 | 0.50 | 0.55 | 0.60 | 0.55 | 0.50 | 0.35 |
| **AVERAGE** | **0.91** | **0.53** | **0.55** | **0.55** | **0.53** | **0.57** | **0.55** | **0.52** | **0.53** |

Table 3: Human accuracy compared to low-level and high-level feature accuracy across all 37 *Close* tasks in Geoclidean.

| AUC | HUMAN | VGG16 LOW | HIGH | RN50 LOW | HIGH | INV3 LOW | HIGH | VIT LOW | HIGH |
|---|---|---|---|---|---|---|---|---|---|
| ANGLE | 0.98 | 0.50 | 0.40 | 0.50 | 0.50 | 0.50 | 0.40 | 0.50 | 0.30 |
| PERP BISECTOR | 0.94 | 0.70 | 0.80 | 0.70 | 0.40 | 0.50 | 0.80 | 0.70 | 0.80 |
| ANG BISECTOR | 0.94 | 0.50 | 0.60 | 0.50 | 0.60 | 0.60 | 0.40 | 0.50 | 0.80 |
| SIXTY ANG | 0.82 | 0.30 | 0.70 | 0.30 | 0.60 | 0.60 | 0.40 | 0.30 | 0.90 |
| RADII | 0.92 | 0.80 | 0.70 | 0.80 | 0.60 | 0.70 | 0.80 | 0.80 | 0.70 |
| DIAMETER | 0.96 | 0.20 | 0.50 | 0.40 | 0.50 | 0.50 | 0.80 | 0.40 | 0.90 |
| SEGMENT | 0.93 | 0.60 | 0.60 | 0.60 | 0.40 | 0.30 | 0.50 | 0.60 | 0.50 |
| RECTILINEAR | 0.90 | 0.60 | 0.40 | 0.60 | 0.40 | 0.50 | 0.60 | 0.70 | 0.40 |
| TRIANGLE | 0.96 | 0.40 | 0.40 | 0.40 | 0.50 | 0.40 | 0.30 | 0.30 | 0.40 |
| QUADRILATERAL | 0.92 | 0.40 | 0.70 | 0.60 | 0.50 | 0.40 | 0.70 | 0.50 | 0.70 |
| EQ T | 0.95 | 0.50 | 0.80 | 0.50 | 0.70 | 0.60 | 0.60 | 0.50 | 0.60 |
| RIGHT ANG T | 0.72 | 0.70 | 0.70 | 0.60 | 0.70 | 0.70 | 0.50 | 0.70 | 0.50 |
| SQUARE | 0.89 | 0.70 | 0.70 | 0.60 | 0.50 | 0.80 | 0.60 | 0.60 | 0.70 |
| RHOMBUS | 0.94 | 0.70 | 0.50 | 0.60 | 0.50 | 0.60 | 0.50 | 0.70 | 0.50 |
| OBLONG | 0.97 | 0.50 | 0.30 | 0.50 | 0.30 | 0.40 | 0.60 | 0.50 | 0.50 |
| RHOMBOID | 0.97 | 0.70 | 0.50 | 0.70 | 0.40 | 0.40 | 0.40 | 0.70 | 0.70 |
| PARALLEL L | 0.95 | 0.60 | 0.20 | 0.60 | 0.30 | 0.40 | 0.40 | 0.40 | 0.70 |
| LLL | 0.97 | 0.50 | 0.60 | 0.50 | 0.10 | 0.40 | 0.30 | 0.50 | 0.70 |
| CLL | 0.95 | 0.30 | 0.50 | 0.60 | 0.30 | 0.30 | 0.60 | 0.50 | 0.60 |
| LLC | 0.68 | 0.70 | 0.40 | 0.60 | 0.50 | 0.40 | 0.50 | 0.60 | 0.50 |
| CCL | 0.87 | 0.60 | 0.60 | 0.50 | 0.70 | 0.30 | 0.60 | 0.50 | 0.70 |
| LCC | 0.89 | 0.50 | 0.60 | 0.50 | 0.60 | 0.70 | 0.60 | 0.40 | 0.70 |
| CCC | 0.67 | 0.60 | 0.50 | 0.60 | 0.60 | 0.60 | 0.60 | 0.60 | 0.90 |
| LLLL | 0.88 | 0.70 | 0.30 | 0.60 | 0.30 | 0.30 | 0.60 | 0.60 | 0.30 |
| LLLC | 0.67 | 0.40 | 0.50 | 0.40 | 0.40 | 0.50 | 0.40 | 0.30 | 0.80 |
| CLLL | 0.84 | 0.60 | 0.70 | 0.60 | 0.60 | 0.70 | 0.50 | 0.70 | 0.60 |
| CLCL | 0.87 | 0.60 | 0.60 | 0.60 | 0.40 | 0.40 | 0.40 | 0.60 | 0.80 |
| LLCC | 0.89 | 0.70 | 0.70 | 0.90 | 0.60 | 0.80 | 0.70 | 0.80 | 0.70 |
| CCCL | 0.92 | 0.30 | 0.60 | 0.30 | 0.60 | 0.30 | 0.70 | 0.30 | 0.70 |
| CLCC | 0.86 | 0.50 | 0.40 | 0.40 | 0.40 | 0.40 | 0.50 | 0.40 | 0.70 |
| CCCC | 0.82 | 0.80 | 0.60 | 0.70 | 0.60 | 0.80 | 0.40 | 0.80 | 1.00 |
| TLL | 0.95 | 0.70 | 0.60 | 0.60 | 0.70 | 0.60 | 0.60 | 0.80 | 0.80 |
| LLT | 0.93 | 0.60 | 0.40 | 0.60 | 0.70 | 0.60 | 0.40 | 0.50 | 0.80 |
| TCL | 0.95 | 0.60 | 0.80 | 0.70 | 0.60 | 0.60 | 0.60 | 0.60 | 0.40 |
| CLT | 0.88 | 0.60 | 0.60 | 0.50 | 0.50 | 0.60 | 0.60 | 0.50 | 0.70 |
| TCC | 0.83 | 0.20 | 0.40 | 0.30 | 0.30 | 0.50 | 0.40 | 0.20 | 0.50 |
| CCT | 0.78 | 0.50 | 0.60 | 0.40 | 0.50 | 0.50 | 0.40 | 0.40 | 0.80 |
| **AVERAGE** | **0.88** | **0.55** | **0.55** | **0.55** | **0.50** | **0.52** | **0.53** | **0.54** | **0.66** |

Table 4: Human accuracy compared to low-level and high-level feature accuracy across all 37 *Far* tasks in Geoclidean.

| AUC | HUMAN | VGG16 | | RN50 | | INV3 | | VIT | |
|---|---|---|---|---|---|---|---|---|---|
| | | LOW | HIGH | LOW | HIGH | LOW | HIGH | LOW | HIGH |
| ANGLE | 0.98 | 0.50 | 0.50 | 0.40 | 0.50 | 0.30 | 0.50 | 0.50 | 0.50 |
| PERP BISECTOR | 0.98 | 0.70 | 0.90 | 0.70 | 0.60 | 0.80 | 1.00 | 0.70 | 0.90 |
| ANG BISECTOR | 0.95 | 0.50 | 0.60 | 0.50 | 0.70 | 0.60 | 0.80 | 0.50 | 0.70 |
| SIXTY ANG | 0.95 | 0.20 | 0.60 | 0.40 | 0.30 | 0.50 | 0.50 | 0.40 | 0.50 |
| RADII | 0.96 | 0.70 | 0.70 | 0.70 | 0.60 | 0.60 | 0.80 | 0.70 | 0.80 |
| DIAMETER | 1.00 | 0.40 | 0.60 | 0.40 | 0.30 | 0.50 | 0.70 | 0.50 | 0.80 |
| SEGMENT | 0.98 | 0.30 | 0.60 | 0.60 | 0.40 | 0.30 | 0.80 | 0.50 | 0.80 |
| RECTILINEAR | 0.90 | 0.70 | 0.30 | 0.60 | 0.50 | 0.60 | 0.60 | 0.60 | 0.50 |
| TRIANGLE | 0.98 | 0.90 | 0.50 | 0.60 | 0.30 | 0.40 | 0.80 | 0.40 | 0.80 |
| QUADRILATERAL | 0.93 | 0.60 | 0.70 | 0.60 | 0.50 | 0.70 | 0.80 | 0.70 | 0.50 |
| EQ T | 0.98 | 0.30 | 0.90 | 0.50 | 0.70 | 0.70 | 0.70 | 0.50 | 0.50 |
| RIGHT ANG T | 0.81 | 0.60 | 0.80 | 0.60 | 0.70 | 0.50 | 0.60 | 0.70 | 0.60 |
| SQUARE | 0.99 | 0.70 | 0.90 | 0.70 | 0.40 | 0.90 | 0.80 | 0.60 | 0.80 |
| RHOMBUS | 0.97 | 0.70 | 0.60 | 0.60 | 0.60 | 0.60 | 0.60 | 0.70 | 0.70 |
| OBLONG | 0.99 | 0.40 | 0.80 | 0.40 | 0.60 | 0.60 | 0.80 | 0.40 | 0.90 |
| RHOMBOID | 0.93 | 0.70 | 0.70 | 0.60 | 0.70 | 0.60 | 0.60 | 0.60 | 0.80 |
| PARALLEL L | 0.96 | 0.50 | 0.50 | 0.40 | 0.50 | 0.50 | 0.70 | 0.50 | 1.00 |
| LLL | 0.97 | 0.50 | 0.80 | 0.60 | 0.70 | 0.70 | 0.40 | 0.60 | 0.90 |
| CLL | 0.97 | 0.20 | 0.70 | 0.20 | 0.30 | 0.30 | 0.60 | 0.10 | 0.60 |
| LLC | 0.92 | 0.60 | 0.70 | 0.60 | 0.60 | 0.70 | 0.80 | 0.60 | 0.70 |
| CCL | 0.88 | 0.70 | 0.40 | 0.60 | 0.60 | 0.30 | 0.60 | 0.60 | 0.80 |
| LCC | 0.96 | 0.40 | 0.80 | 0.40 | 0.80 | 0.60 | 0.80 | 0.50 | 1.00 |
| CCC | 0.88 | 0.60 | 0.60 | 0.60 | 0.80 | 0.80 | 0.90 | 0.60 | 0.90 |
| LLLL | 0.98 | 0.50 | 0.30 | 0.40 | 0.30 | 0.30 | 0.40 | 0.40 | 0.70 |
| LLLC | 0.89 | 0.20 | 0.80 | 0.40 | 0.70 | 0.50 | 0.70 | 0.30 | 0.80 |
| CLLL | 0.90 | 0.50 | 0.60 | 0.60 | 0.70 | 0.60 | 0.50 | 0.70 | 0.70 |
| CLCL | 0.86 | 0.60 | 0.70 | 0.70 | 0.50 | 0.40 | 0.30 | 0.60 | 0.90 |
| LLCC | 0.93 | 0.80 | 0.60 | 0.80 | 0.50 | 0.80 | 0.70 | 0.80 | 0.70 |
| CCCL | 0.93 | 0.40 | 0.60 | 0.70 | 0.70 | 0.40 | 0.60 | 0.60 | 0.80 |
| CLCC | 0.90 | 0.70 | 0.70 | 0.70 | 0.80 | 0.80 | 1.00 | 0.70 | 0.90 |
| CCCC | 0.88 | 0.80 | 0.70 | 0.70 | 0.60 | 0.70 | 0.60 | 0.70 | 0.90 |
| TLL | 0.98 | 0.70 | 0.70 | 0.70 | 0.50 | 0.60 | 0.50 | 0.70 | 0.90 |
| LLT | 0.94 | 0.50 | 0.40 | 0.50 | 0.60 | 0.50 | 0.90 | 0.50 | 0.90 |
| TCL | 0.96 | 0.70 | 0.60 | 0.70 | 0.50 | 0.60 | 0.60 | 0.70 | 0.50 |
| CLT | 0.89 | 0.40 | 0.60 | 0.40 | 0.50 | 0.50 | 0.70 | 0.40 | 0.70 |
| TCC | 0.85 | 0.40 | 0.50 | 0.40 | 0.40 | 0.60 | 0.50 | 0.40 | 0.40 |
| CCT | 0.80 | 0.60 | 0.80 | 0.50 | 0.70 | 0.70 | 0.70 | 0.60 | 0.90 |
| **AVERAGE** | **0.93** | **0.55** | **0.64** | **0.55** | **0.56** | **0.57** | **0.67** | **0.56** | **0.75** |

Table 5: Human AUC compared to low-level and high-level feature AUC across all 37 *Close* tasks in Geoclidean.

| AUC | HUMAN | VGG16 LOW | HIGH | RN50 LOW | HIGH | INV3 LOW | HIGH | VIT LOW | HIGH |
|---|---|---|---|---|---|---|---|---|---|
| ANGLE | 1.00 | 0.60 | 0.44 | 0.48 | 0.40 | 0.60 | 0.20 | 0.52 | 0.16 |
| PERP BISECTOR | 1.00 | 0.52 | 0.80 | 0.52 | 0.28 | 0.52 | 0.96 | 0.52 | 0.88 |
| ANG BISECTOR | 1.00 | 0.32 | 0.64 | 0.36 | 0.68 | 0.48 | 0.52 | 0.36 | 0.88 |
| SIXTY ANG | 1.00 | 0.32 | 0.60 | 0.32 | 0.44 | 0.36 | 0.52 | 0.36 | 1.00 |
| RADII | 1.00 | 0.76 | 0.84 | 0.76 | 0.56 | 0.76 | 0.80 | 0.72 | 0.60 |
| DIAMETER | 1.00 | 0.00 | 0.76 | 0.00 | 0.48 | 0.32 | 0.84 | 0.16 | 1.00 |
| SEGMENT | 1.00 | 0.48 | 0.64 | 0.52 | 0.44 | 0.48 | 0.48 | 0.40 | 0.72 |
| RECTILINEAR | 1.00 | 0.76 | 0.40 | 0.68 | 0.52 | 0.56 | 0.36 | 0.52 | 0.32 |
| TRIANGLE | 1.00 | 0.20 | 0.40 | 0.28 | 0.36 | 0.28 | 0.48 | 0.08 | 0.36 |
| QUADRILATERAL | 1.00 | 0.36 | 0.52 | 0.32 | 0.44 | 0.36 | 0.56 | 0.36 | 0.72 |
| EQ T | 1.00 | 0.44 | 0.80 | 0.68 | 0.88 | 0.72 | 0.88 | 0.68 | 0.68 |
| RIGHT ANG T | 0.94 | 0.76 | 0.80 | 0.76 | 0.64 | 0.80 | 0.64 | 0.80 | 0.64 |
| SQUARE | 1.00 | 0.64 | 0.80 | 0.80 | 0.56 | 0.88 | 0.88 | 1.00 | 0.64 |
| RHOMBUS | 1.00 | 0.40 | 0.40 | 0.56 | 0.44 | 0.76 | 0.48 | 0.60 | 0.56 |
| OBLONG | 1.00 | 0.56 | 0.08 | 0.20 | 0.24 | 0.16 | 0.52 | 0.32 | 0.52 |
| RHOMBOID | 1.00 | 0.76 | 0.56 | 0.64 | 0.28 | 0.48 | 0.60 | 0.64 | 0.72 |
| PARALLEL L | 1.00 | 0.52 | 0.32 | 0.52 | 0.16 | 0.32 | 0.36 | 0.40 | 0.92 |
| LLL | 1.00 | 0.20 | 0.56 | 0.24 | 0.08 | 0.28 | 0.32 | 0.28 | 0.76 |
| CLL | 1.00 | 0.32 | 0.60 | 0.24 | 0.12 | 0.20 | 0.36 | 0.20 | 0.76 |
| LLC | 1.00 | 0.48 | 0.28 | 0.44 | 0.56 | 0.36 | 0.44 | 0.36 | 0.32 |
| CCL | 1.00 | 0.56 | 0.64 | 0.56 | 0.68 | 0.40 | 0.72 | 0.56 | 0.80 |
| LCC | 1.00 | 0.56 | 0.60 | 0.64 | 0.52 | 0.68 | 0.72 | 0.60 | 1.00 |
| CCC | 1.00 | 0.64 | 0.20 | 0.92 | 0.76 | 0.80 | 0.64 | 0.92 | 1.00 |
| LLLL | 1.00 | 0.72 | 0.20 | 0.72 | 0.16 | 0.40 | 0.88 | 0.68 | 0.20 |
| LLLC | 1.00 | 0.28 | 0.56 | 0.32 | 0.64 | 0.40 | 0.60 | 0.28 | 0.76 |
| CLLL | 1.00 | 0.76 | 0.76 | 0.64 | 0.92 | 0.72 | 0.60 | 0.72 | 0.52 |
| CLCL | 1.00 | 0.48 | 0.52 | 0.44 | 0.52 | 0.60 | 0.52 | 0.44 | 0.92 |
| LLCC | 1.00 | 0.84 | 0.76 | 0.88 | 0.80 | 0.80 | 0.68 | 0.76 | 0.88 |
| CCCL | 1.00 | 0.20 | 0.36 | 0.32 | 0.72 | 0.36 | 0.60 | 0.32 | 0.88 |
| CLCC | 1.00 | 0.48 | 0.32 | 0.40 | 0.52 | 0.36 | 0.64 | 0.44 | 0.88 |
| CCCC | 1.00 | 0.80 | 0.76 | 0.72 | 0.64 | 0.80 | 0.32 | 0.84 | 1.00 |
| TLL | 1.00 | 0.76 | 0.60 | 0.72 | 0.68 | 0.72 | 0.80 | 0.76 | 0.80 |
| LLT | 1.00 | 0.60 | 0.48 | 0.64 | 0.64 | 0.40 | 0.48 | 0.68 | 0.80 |
| TCL | 1.00 | 0.64 | 0.76 | 0.72 | 0.64 | 0.72 | 0.48 | 0.80 | 0.56 |
| CLT | 1.00 | 0.48 | 0.68 | 0.48 | 0.32 | 0.64 | 0.80 | 0.68 | 0.72 |
| TCC | 1.00 | 0.16 | 0.20 | 0.20 | 0.08 | 0.24 | 0.32 | 0.20 | 0.36 |
| CCT | 1.00 | 0.44 | 0.68 | 0.48 | 0.44 | 0.48 | 0.52 | 0.48 | 0.92 |
| **AVERAGE** | **1.00** | **0.51** | **0.55** | **0.52** | **0.49** | **0.52** | **0.58** | **0.53** | **0.71** |

Table 6: Human AUC compared to low-level and high-level feature AUC across all 37 *Far* tasks in Geoclidean.

| AUC | HUMAN | VGG16 | | RN50 | | INV3 | | VIT | |
|---|---|---|---|---|---|---|---|---|---|
| | | LOW | HIGH | LOW | HIGH | LOW | HIGH | LOW | HIGH |
| ANGLE | 1.00 | 0.32 | 0.56 | 0.28 | 0.48 | 0.32 | 0.40 | 0.32 | 0.48 |
| PERP BISECTOR | 1.00 | 0.76 | 0.92 | 1.00 | 0.60 | 0.88 | 1.00 | 0.56 | 0.84 |
| ANG BISECTOR | 1.00 | 0.36 | 0.60 | 0.40 | 0.60 | 0.52 | 0.72 | 0.36 | 0.80 |
| SIXTY ANG | 1.00 | 0.16 | 0.48 | 0.16 | 0.28 | 0.44 | 0.48 | 0.32 | 0.28 |
| RADII | 1.00 | 0.60 | 0.72 | 0.60 | 0.60 | 0.56 | 0.68 | 0.56 | 0.68 |
| DIAMETER | 1.00 | 0.00 | 0.88 | 0.24 | 0.16 | 0.48 | 0.68 | 0.24 | 1.00 |
| SEGMENT | 1.00 | 0.24 | 0.80 | 0.28 | 0.16 | 0.36 | 0.80 | 0.36 | 1.00 |
| RECTILINEAR | 1.00 | 0.84 | 0.32 | 0.64 | 0.68 | 0.36 | 0.48 | 0.64 | 0.72 |
| TRIANGLE | 1.00 | 0.80 | 0.52 | 0.76 | 0.40 | 0.44 | 0.84 | 0.76 | 0.80 |
| QUADRILATERAL | 1.00 | 0.72 | 0.60 | 0.68 | 0.44 | 0.72 | 0.76 | 0.68 | 0.56 |
| EQ T | 1.00 | 0.36 | 0.88 | 0.68 | 0.84 | 0.84 | 0.72 | 0.60 | 0.44 |
| RIGHT ANG T | 1.00 | 0.52 | 0.88 | 0.64 | 0.64 | 0.52 | 0.76 | 0.52 | 0.88 |
| SQUARE | 1.00 | 0.84 | 0.88 | 0.92 | 0.64 | 0.88 | 0.92 | 0.88 | 0.96 |
| RHOMBUS | 1.00 | 0.52 | 0.56 | 0.72 | 0.44 | 0.76 | 0.64 | 0.84 | 0.76 |
| OBLONG | 1.00 | 0.64 | 0.88 | 0.60 | 0.52 | 0.76 | 1.00 | 0.56 | 1.00 |
| RHOMBOID | 1.00 | 0.68 | 0.72 | 0.68 | 0.60 | 0.72 | 0.64 | 0.76 | 0.96 |
| PARALLEL L | 1.00 | 0.44 | 0.52 | 0.44 | 0.40 | 0.40 | 0.64 | 0.40 | 1.00 |
| LLL | 1.00 | 0.56 | 0.72 | 0.68 | 0.52 | 0.68 | 0.32 | 0.64 | 1.00 |
| CLL | 1.00 | 0.04 | 0.56 | 0.20 | 0.00 | 0.20 | 0.64 | 0.12 | 1.00 |
| LLC | 1.00 | 0.52 | 0.68 | 0.60 | 0.60 | 0.76 | 0.88 | 0.52 | 0.68 |
| CCL | 1.00 | 0.60 | 0.48 | 0.56 | 0.60 | 0.36 | 0.52 | 0.48 | 0.96 |
| LCC | 1.00 | 0.64 | 0.84 | 0.60 | 0.80 | 0.72 | 0.88 | 0.60 | 1.00 |
| CCC | 1.00 | 0.68 | 0.72 | 0.76 | 0.92 | 0.68 | 0.96 | 0.76 | 0.96 |
| LLLL | 1.00 | 0.56 | 0.32 | 0.56 | 0.28 | 0.36 | 0.36 | 0.56 | 0.64 |
| LLLC | 1.00 | 0.16 | 0.72 | 0.32 | 0.84 | 0.56 | 0.80 | 0.20 | 0.84 |
| CLLL | 1.00 | 0.56 | 0.64 | 0.52 | 0.56 | 0.48 | 0.72 | 0.60 | 0.68 |
| CLCL | 1.00 | 0.48 | 0.72 | 0.52 | 0.40 | 0.60 | 0.48 | 0.40 | 0.92 |
| LLCC | 1.00 | 0.84 | 0.60 | 0.72 | 0.64 | 0.64 | 0.60 | 0.80 | 0.80 |
| CCCL | 1.00 | 0.36 | 0.68 | 0.48 | 0.76 | 0.56 | 0.80 | 0.52 | 0.96 |
| CLCC | 1.00 | 0.76 | 0.84 | 0.88 | 0.84 | 0.96 | 1.00 | 0.88 | 0.88 |
| CCCC | 1.00 | 0.84 | 0.68 | 0.96 | 0.44 | 0.92 | 0.68 | 0.96 | 0.92 |
| TLL | 1.00 | 0.68 | 0.72 | 0.68 | 0.60 | 0.76 | 0.64 | 0.64 | 0.80 |
| LLT | 1.00 | 0.52 | 0.32 | 0.56 | 0.60 | 0.44 | 0.88 | 0.52 | 0.84 |
| TCL | 1.00 | 0.64 | 0.52 | 0.64 | 0.36 | 0.60 | 0.40 | 0.68 | 0.56 |
| CLT | 1.00 | 0.28 | 0.64 | 0.48 | 0.28 | 0.68 | 0.88 | 0.32 | 0.72 |
| TCC | 1.00 | 0.36 | 0.48 | 0.48 | 0.36 | 0.48 | 0.52 | 0.44 | 0.44 |
| CCT | 1.00 | 0.52 | 0.88 | 0.56 | 0.76 | 0.60 | 0.76 | 0.64 | 0.96 |
| **AVERAGE** | **1.00** | **0.52** | **0.66** | **0.58** | **0.53** | **0.59** | **0.70** | **0.56** | **0.80** |

### A.8 Datasheet for Geoclidean

**Motivation.**

1. *For what purpose was the dataset created?* Geoclidean was created to study few-shot generalization of humans and machine in Euclidean geometry.

2. *Who created the dataset (e.g., which team, research group) and on behalf of which entity (e.g., company, institution, organization)?* Stanford AI Lab.

3. *Who funded the creation of the dataset?* Stanford AI Lab.

**Composition.**

1. *What do the instances that comprise the dataset represent (e.g., documents, photos, people, countries)?* Images of lines and circles procedurally generated by the Geoclidean framework, each one representing an Euclidean geometry concept.

2. *How many instances are there in total (of each type, if appropriate)?* There are 740 images total.

3. *Does the dataset contain all possible instances or is it a sample (not necessarily random) of instances from a larger set?* Geoclidean can generate infinitely many concepts and images from the Euclidean geometry universe, and hence the datasets we provide are a sample.

4. *What data does each instance consist of?* Images of lines and circles.

5. *Is there a label or target associated with each instance?* The label for each image is the geometric concept from which it was rendered from.

6. *Is the dataset self-contained, or does it link to or otherwise rely on external resources (e.g., websites, tweets, other datasets)?* The dataset is self-contained.

**Collection Process.**

1. *How was the data associated with each instance acquired?* The data was procedurally generated with Geoclidean; framework to create the dataset is released as well.

2. *If the dataset is a sample from a larger set, what was the sampling strategy (e.g., deterministic, probabilistic with specific sampling probabilities)?* We choose Euclidean geometry concepts from definitions in the first book of Euclid's Elements [Simson et al., 1838] as well as from concepts with possible orderings of three, four, and five objects.

**Preprocessing/cleaning/labeling.**

1. *Was any preprocessing/cleaning/labeling of the data done (e.g., discretization or bucketing, tokenization, part-of-speech tagging, SIFT feature extraction, removal of instances, processing of missing values)?* N/A.

2. *Was the "raw" data saved in addition to the preprocessed/cleaned/labeled data (e.g., to support unanticipated future uses)?* Only raw data is provided.

3. *Is the software used to preprocess/clean/label the instances available?* N/A.

**Uses.**

1. *Has the dataset been used for any tasks already?* The dataset has been used for the human experiments and model benchmarking reported in the main text of the paper.

2. *Is there anything about the composition of the dataset or the way it was collected and preprocessed/cleaned/labeled that might impact future uses?* No.

**Distribution.**

1. *Will the dataset be distributed to third parties outside of the entity (e.g., company, institution, organization) on behalf of which the dataset was created?* The dataset is publicly released.

2. *How will the dataset will be distributed (e.g., tarball on website, API, GitHub)?* Geoclidean is hosted on the Stanford CS cluster.

3. *When will the dataset be distributed?* The dataset is available for download.

4. *Will the dataset be distributed under a copyright or other intellectual property (IP) license, and/or under applicable terms of use (ToU)?* Geoclidean is licensed under CC-BY 4.0.

5. *Have any third parties imposed IP-based or other restrictions on the data associated with the instances?* No.

**Maintenance.**

1. *Who will be supporting/hosting/maintaining the dataset?* The members of Stanford AI Lab.

2. *How can the owner/curator/manager of the dataset be contacted (e.g., email address)?* Authors of this paper can be contacted via the provided emails.

3. *Will the dataset be updated (e.g., to correct labeling errors, add new instances, delete instances)?* No.

4. *If others want to extend/augment/build on/contribute to the dataset, is there a mechanism for them to do so?* New image datasets of Euclidean geometry concepts may be created with the Geoclidean framework, but not added to our released dataset.