# OpenReview forum: "Geoclidean: Few-Shot Generalization in Euclidean Geometry"
_NeurIPS.cc/2022/Track/Datasets_and_Benchmarks — NeurIPS 2022 Datasets and Benchmarks _

### Official Review · Reviewer_qXzh · 2022-07-18
**Novel Dataset and Benchmark that exposes the gap between human and pretrained vision performance**

**Rating:** 7
**Confidence:** 4
**Clarity:** Yes. But please add description of wh…

**Strengths:**

- The language to generate samples from the dataset is particularly useful to further extend this research direction
- The author poses an interesting question through the benchmark task: can a vision model trained on natural images perform well on few-shot generalization of geometric euclidean concept? The comparison with human performance further strengthens the importance of this task.
- The paper is very well-written. A reader with minimal geometry concept learning background should easily get the gist of the paper.


**Weaknesses:**

1.  Although the author clearly shows this task's importance, the reviewer still questions whether evaluating the pre-trained model for this task is fair. Perhaps this point can be clarified by citing literature that shows a strong relation between geometric concepts and downstream tasks.
2. It is unclear what are low-level and high-level features? Adding one-liner description of these will be useful.
3. What is the y-axis of the histogram for signifies? Is it the percentage of tasks with x% accuracy? Perhaps clarifying these will be useful, since the histograms are the main visualizations in this paper.

**Additional Feedback:**

Typo in line 106. "we rejection sample realization" should be "we reject sample realization"

**Correctness:**

Dataset: Claims are correct and the dataset is constructed in in a sound way.
Benchmark: The paper is of sound evaluation methods and experiment design. Although adding citation mentioned in the 1st point in the weakness, or maybe adding another evaluation to prove that can further strengthen this paper.

**Documentation:**

The dataset is well-documented

**Ethics:**

There is no ethical concern

**Relation To Prior Work:**

Yes.

**Summary And Contributions:**

The authors present the following:
- Two datasets and a benchmark task that test the generalization capacity of human and vision model on the Euclidean geometry concept.
- Language to generate versions of the presented dataset flexibly.
- Evaluation of human and pretrained model performances on the datasets
- The benchmark task is a few-shot generalization of positive images from specific concepts (e.g., a circle inside a triangle).

---

> ### Author Response · Authors · 2022-08-14
> **Response to Reviewer QXZH pt 1**
>
> We thank you for the constructive feedback!
>
> Q: Fairness of evaluation on pretrained models and relation between geometric concepts and downstream tasks.
>
> A: We thank you for bringing up this point. Many downstream tasks that may leverage pretrained vision models would greatly benefit from geometric reasoning, such as construction (LegoTron [1], Physical Construction Tasks [2]), physical reasoning (CLEVRER [3], ThreeDWorld [4]), and shape understanding tasks (PartNet [5], ShapeNet [6]). We thank you for the suggestion and have highlighted these examples in the main text (Section 6 Discussion) and in the Appendix (A.7 Discussion).
>
> We would also like to clarify that our goal in this paper is to present a task that tests a core component of human vision; humans successfully achieve high accuracy on the Geoclidean tasks even if most training exposure is to natural images. Indeed, Kennedy and Ross [7] showed that humans without significant exposure to pictorial art are able to identify a wide range of outline drawings. We believe it is an interesting task to evaluate whether pretrained vision models are intrinsically sensitive to Euclidean geometry, as humans are.
>
> As suggested by reviewer bmE1, we also added an evaluation of vision models pretrained on the Google Quick Draw dataset [8], which contain images with line drawings. This dataset contains images more similar to the rendered images of Geoclidean, making for a more fair comparison in terms of smaller visual distribution shift. We have trained the VGG16, ResNet50, InceptionV3, and Vision Transformer model on the Google Quick Draw dataset, and tested them on all Geoclidean tasks. On the Quick Draw dataset, these models achieved a top-1 candidate test accuracy of 0.5455, 0.6752, 0.6921, and 0.7039, respectively. These values are in the same range as the reported value of approximately 0.7 for the released recurrent neural network in the official Github repository [9].
>
> Below, we report the average accuracy for Quick Draw–pretrained models across all Geoclidean tasks, for each vision model and for low-level and high-level features, compared to the ImageNet pretrained models. We see that ImageNet pretrained models are comparable with Quick Draw–pretrained models in low-level features, and outperform Quick Draw–pretrained models in high-level features. We hypothesize that though the visual domain shift from Quick Draw to Geoclidean is smaller than that of ImageNet to Geoclidean, there is added complexity of geometric reasoning in our proposed tasks that is not captured by drawing classification. Hence, this illustrates precisely what we aim to test with Geoclidean.
>
> |                       | Human  | VGG16  |      | RN50  |      | INV3  |      | VIT  |      |
> | -----------           | ------ | ------ | -----| ------| -----| ------| -----| ------| -----|
> |                       |        | Low    | High | Low    | High | Low    | High | Low    | High |
> | ImageNet   pretrained | 0.91   | 0.55   | 0.60 | 0.55  | 0.53 | 0.54  | 0.60 | 0.55  | 0.70 |
> | Quick Draw pretrained | 0.91   | 0.52   | 0.54 | 0.55  | 0.53 | 0.56  | 0.55 | 0.55  | 0.57 |
>
> We have added results from Quick Draw pretrained models with accuracy for each task to the Appendix (A.5 Pretraining).
>
> [1] Walsman, A., Zhang, M., Fishman, A., Desingh, K., Fox, D., & Farhadi, A. LegoTron: An Environment for Interactive Structural Understanding.
>
> [2] Bapst, V., Sanchez-Gonzalez, A., Doersch, C., Stachenfeld, K., Kohli, P., Battaglia, P., & Hamrick, J. (2019, May). Structured agents for physical construction. In International conference on machine learning (pp. 464-474). PMLR.
>
> [3] Yi, K., Gan, C., Li, Y., Kohli, P., Wu, J., Torralba, A., & Tenenbaum, J. B. (2019). Clevrer: Collision events for video representation and reasoning. ICLR 2020.
>
> [4] Gan, C., Schwartz, J., Alter, S., Schrimpf, M., Traer, J., De Freitas, J., ... & Yamins, D. L. (2020). Threedworld: A platform for interactive multi-modal physical simulation. NeurIPS Dataset 2021.
>
> [5] Mo, K., Zhu, S., Chang, A. X., Yi, L., Tripathi, S., Guibas, L. J., & Su, H. (2019). Partnet: A large-scale benchmark for fine-grained and hierarchical part-level 3d object understanding. In Proceedings of the IEEE/CVF conference on computer vision and pattern recognition (pp. 909-918).
>
> [6] Chang, A. X., Funkhouser, T., Guibas, L., Hanrahan, P., Huang, Q., Li, Z., ... & Yu, F. (2015). Shapenet: An information-rich 3d model repository. arXiv preprint arXiv:1512.03012.
>
> [7] Kennedy, J. M., & Ross, A. S. (1975). Outline picture perception by the Songe of Papua. Perception, 4(4), 391-406.
>
> [8] Ha, D., & Eck, D. (2017). A neural representation of sketch drawings. ICLR 2018.
>
> [9] https://github.com/tensorflow/docs/blob/master/site/en/r1/tutorials/sequences/recurrent_quickdraw.md

---

> > ### Author Response · Authors · 2022-08-14
> > **Response to Reviewer QXZH pt 2**
> >
> > Q: Description of low and high-level features.
> >
> > A: Low-level features are outputs of earlier layers in neural networks, which tend to capture low-level information such as edges and primitive shapes, while high-level features are outputs of later layers that tend to capture more high-level semantic information [10]. We detail how we define the layers for each baseline in the Appendix (A.6 Model Benchmarks), and have added descriptions for the features in the main text.
> >
> > We have also included feature visualizations of low and high-level features in the Appendix (A.4 Feature Visualizations) for additional analyses, comparing Geoclidean tasks that require reasoning, to perception tasks involving simple geometric primitives and perturbations. These comparisons highlight Geoclidean as a unique and interesting test for vision models.
> >
> > [10] Zeiler, M. D., & Fergus, R. (2014, September). Visualizing and understanding convolutional networks. In European conference on computer vision (pp. 818-833). Springer, Cham.
> >
> > --------
> >
> > Q: Y-axis for histograms.
> >
> > A: The y-axis of the histograms indeed signifies the percentage of tasks with x% accuracy. We apologize for the confusion and have clarified this in the main text.
> >
> > --------
> >
> > Typo: We thank you for the comment and have edited the main text!
> >
> > --------
> >
> > We have highlighted all changes made in the main text and Appendix in red for clarity; we hope we have addressed your concerns, and would be happy to further discuss.

---

### Official Review · Reviewer_yPZW · 2022-07-19
**Euclidean database for image generation**

**Rating:** 8
**Confidence:** 3

**Strengths:**

Comparison of human accuracy with computer vision accuracy to lay a suitable baseline
Evaluation of the Euclidean geometry with computer vision models
Introduction of a new language, which can render various geometric pattern, which then makes the dataset suitable for Few shot learning task, as well as extending it for educational purposes and maybe extension into n-dimensions

**Weaknesses:**

Some qualitative examples of the experiments would have been nice as well as bigger fonts in the figures, which makes the figures hard to read, when zooming in.


**Additional Feedback:**

I don't have anything to add here, except what was mentioned in the weaknesses.

**Clarity:**

The paper was written clearly, as I could understand and follow it.


**Correctness:**

To my knowledge the methods seem to be correct


**Documentation:**

The dataset is well documented, as the GitHub link was easy to find in the supplementary materials. Also the data-sheet provides sufficient information.


**Ethics:**

The ethics have been discussed and also the crowdsourced data was used in an ethical way.


**Relation To Prior Work:**

All related work with geometric datasets and few shot learning have been mentioned.


**Summary And Contributions:**

Euclidean geometry is an old mathematical branch. Even though those shapes do not exits in the real world humans are good in recognising those patterns. The paper tries to answer if machine learning models can be equally good at it. To answer the question the paper introduces a euclidean geometry shape dataset as well as a language to create those shapes. Also a novel few-shot generalisation benchmark is introduced to evaluate computer vision models on this dataset.

---

> ### Author Response · Authors · 2022-08-14
> **Response to Reviewer YPZW**
>
> We thank you for the constructive feedback! We have edited our figures with bigger fonts in the main text. We have also included qualitative examples of experiments in the Appendix (A.3 Human Experiments), and in the provided dataset (A.2 Data).

---

### Official Review · Reviewer_TsFh · 2022-07-27
**Interesting idea and apt framework**

**Rating:** 8
**Confidence:** 4
**Correctness:** Everything seems correct.
**Clarity:** The paper is well-written.

**Strengths:**

The idea is interesting, the gap between humans and AI models convincing, while the framework seems apt for benchmarking attempts to close this gap.

**Weaknesses:**

By giving more (than just 5) examples to the AI models, an exploration of how much the gap could be closed between humans and AI models would be interesting, but it is not necessary


**Additional Feedback:**

N/A

**Documentation:**

Documentation seems sufficient.

**Ethics:**

No ethical concerns seem to be present.

**Relation To Prior Work:**

This is not my area of research, but the authors seem to cite and discuss prior work.

**Summary And Contributions:**

The authors propose a framework for Euclidean geometry: a language for representing anything which can be drawn with a straight edge and a compass.  They then show that humans are able to generalize such concepts after few examples, while state-of-the-art computer vision models perform considerably less well.  The idea is interesting, the gap between humans and AI models convincing, while the framework seems apt for benchmarking attempts to close this gap.  I only have a few minor comments

- p. 4, line 106: "we rejection sample realizations" -> "we reject sample realizations"

- p. 6, I think I understand what is trying to be conveyed here, however the paragraph starting on line 197 is a bit confusing with "each of 10 test images of 5 positive examples and 5 negative examples from either Close or Far..." when the paragraph above mentions 15 test images of 5 positive examples, 5 negative examples of Close, and 5 negative examples of Far

- p. 9, line 309: "negative society impact" -> "negative societal impact"

---

> ### Author Response · Authors · 2022-08-14
> **Response to Reviewer TSFH**
>
> We thank you for the constructive feedback! We have edited the main text to address minor comments. We agree that an exploration of the gap between humans and AI models is interesting, and can be conducted with the Geoclidean framework we released as a future research direction.

---

### Official Review · Reviewer_V6yt · 2022-07-27
**Good contribution towards the vision problems involving plane geometry**

**Rating:** 6
**Confidence:** 4
**Correctness:** It is constructed in a sound way.
**Clarity:** Yes. It is well written.

**Strengths:**

1. This paper conducted solid works on a less explored problem, which is related to the basic challenges and limitations of existing computer vision models.

2. The developed library can be used in various follow-up works.

3. The gained insights like the gap between humans and computer vision models, the difference in performance between CNN-based models and ViT are inspiring.

**Weaknesses:**

For it is finite in my knowledge, I think it can be problematic to compare humans and vision models using the task and setting in the paper, as well as using the few-shot generalization task to evaluate the human-like visual competencies of vision models. The reasons are as follows:
1. The humans actually conduct perception and reasoning (with Euclidean geometry knowledge and intuition)  processes while the vision models only conduct the perception (with representations from pretrained models) process. One possible supplementary can be adding experiments of humans with basic or no Euclidean geometry knowledge ( like elementary school students or infants).

2. If the target of the proposed benchmark is to evaluate the capability in Euclidean geometry concept learning or sensitivity to Euclidean geometry of different vision models, I think there can be better evaluating tasks rather than few-shot generalization task, for example, involving NLP models to discriminate whehter the concept and rendered image are corresponding or not, which can be trained on a generated large-scale dataset, rather than no training or fine-tuning. If there are follow-up research works on it, I think Capsule Networks can be a competitor on this benchmark.

3. The threshold is an adaptive value. I think it can be more reasonable to add the results by using the same models with initialized parameters, to compare with the results by pretrained models.

**Additional Feedback:**

No.

**Documentation:**

Yes.

**Ethics:**

No.

**Relation To Prior Work:**

Yes. It is clearly discussed how this work differs from previous contributions.

**Summary And Contributions:**

This paper proposes a domain-specific language for the Euclidean concept realization and implemented rendering library, as well as two datasets for the few-shot concept learning task, which is used to evaluate the generalization capabilities of humans and pretrained deep learning models. The rendering library and datasets are good contributions to the plane geometry problems in computer vision. The experiments conducted show the gap in performance between humans and deep learning models, which is a good start for follow-up explorations to fix this gap.

---

> ### Author Response · Authors · 2022-08-14
> **Response to Reviewer V6YT**
>
> We thank you for the constructive feedback!
>
> Q: Comparison of human and vision models.
>
> A: We thank you for bringing up the great point that humans are engaged in perception and reasoning while models are not. We agree. Though unfortunately we don’t have the capability to easily test elementary school students or infants (or perhaps an even better comparison, to do speeded judgment from brief visual presentation), the very fact that adult humans likely use reasoning to solve our task highlights an important contribution: it will allow for better testing of vision models that do aim to incorporate reasoning and high level semantics. The community is moving from using deep learning for perception only to integrated perception and reasoning. This is why we believe it’s important to contribute tasks like Geoclidean that can verify this in the geometric reasoning domain.
>
> Given feedback, we have also added additional feature visualizations in the Appendix (A.4 Feature Visualizations), comparing Geoclidean tasks that require reasoning, to perception tasks involving simple geometric primitives and perturbations. These comparisons highlight Geoclidean as a unique and interesting test for vision models.
>
> --------
>
> Q: Evaluation task.
>
> A: We agree that there are many interesting evaluation tasks that can be built on the Geoclidean framework, such as those involving language models applied to the Geoclidean DSL, or those involving training on a generated large-scale dataset. We propose evaluation on the few-shot generalization task, as the simplest approach to investigating the ability of pretrained vision models to capture intrinsic geometric properties as humans do -- can features captured by vision models generalize image classes according to Euclidean geometry? We are certainly excited about follow-up research directions that are enabled by the Geoclidean DSL, such as using the Geoclidean framework to generate instances that cover geometrically equivalent concepts, or bridging to natural language. We thank you for bringing up these discussion points, and have added a section (on this point and the above) to the main text (Section 6 Discussion) and Appendix (A.7 Discussion).
>
> --------
>
> Q: Initialized and pretrained model comparison.
>
> A: We thank you for the suggestion to compare randomly initialized vision models with ImageNet pretrained models. Below, we report the average accuracy for both across all Geoclidean tasks, for each vision model and for low-level and high-level features. We see that high-level features from ImageNet pretrained models consistently outperform features from the random models. We hypothesize that this is due to the higher-level semantic information encoded in later layers that capture some capability to conduct geometric reasoning. We have added results from randomly initialized models with accuracy for each task to the Appendix (A.5 Pretraining).
>
> |                       | Human  | VGG16  |      | RN50  |      | INV3  |      | VIT  |      |
> | -----------           | ------ | ------ | -----| ------| -----| ------| -----| ------| -----|
> |                       |        | Low    | High | Low    | High | Low    | High | Low    | High |
> | ImageNet   pretrained | 0.91   | 0.55   | 0.60 | 0.55  | 0.53 | 0.54  | 0.60 | 0.55  | 0.70 |
> | Not pretrained        | 0.91   | 0.53   | 0.55 | 0.55  | 0.53 | 0.57  | 0.55 | 0.52  | 0.53 |
>
> --------
>
> We have highlighted all changes made in the main text and Appendix in red for clarity; we hope we have addressed your concerns, and would be happy to further discuss.

---

### Official Review · Reviewer_bmE1 · 2022-07-28
**Useful dataset with lots of potential applications**

**Rating:** 7
**Confidence:** 3
**Clarity:** The paper is well-written and easy to…

**Strengths:**

1. It introduces a very fundamental task in image understanding. Geometric shapes appear everywhere, and having the ability to generate complex shapes easily will be very useful for the community.

2. Employs novel near/far categorizations to comprehend the nuances of geometry. This will enable debugging future models and architectures for geometric patterns and help build models that are robust to geometric object perturbations like rotations. This will also enable research in disentanglement to understand specific geometric patterns that models learn and patterns that models struggle with.

EDIT: The authors addressed nearly all of the concerns and performed additional experiments to validate the claims. This dataset would be highly beneficial to the community and I have updated the ratings.

**Weaknesses:**

1. The motivation to just use models trained on natural images is somewhat unclear. Deep learning models are domain specific and they often don’t generalize well to other domains outside the training data. For example, Berry et al. (1) changed the background of images, and that reduced the performance of the models. So it would be great to have a clearer description of why the authors think that these vision models should understand geometric objects. Thus, is this an appropriate goal to test the few-shot generalization of existing vision models trained on natural images?

2. Using the low-level features from pre-trained models to evaluate if models do learn some low-level features was an interesting experiment, but I would have loved some comparison and analysis on this part. Do vision transformers perform better than resnet models at low-level feature understanding with respect to geometric objects? Are some geometric structures too complex for these low-level features, like two circles and one line? What is a simple structure (like a circle or a line) for the model to generalize? This would also help illustrate newer insights into low-level features learned by these models. Since the vision transformers worked quite well compared to the other models, the low-level feature comparisons would be very insightful.

3. The model does near-far categorization. But there isn’t much analysis or insight into this category. What was the conclusion of the learned manifold? How robust are the learned images to near-versus-far-categorizations to simple perturbations in geometry (like angle distortions or scale)? In other words, do these models actually learn the near and far categorizations well from the geometric perspective?

**Additional Feedback:**

I think the authors could make a stronger case by showing that training a model specific to these geometric images and evaluating if models do learn these geometric concepts. If the goal is to understand geometry why not build a model only to understand this concept?

The few-shot generalization using a dataset like google draw could provide useful insights into the few-shot generalizability to geometric objects.



**Correctness:**

I think the authors could make a stronger case by showing that training a model specific to these geometric images and evaluating if models do learn these geometric concepts. If the goal is to understand geometry why not build a model only to understand this concept? The few-shot reasoning to understand geometry by using vision models is not very intuitive to me because models do not generalize well to out of domain images as mentioned above (1). Even training on something like Google Draw dataset (https://github.com/googlecreativelab/quickdraw-dataset) could provide some interesting insights. Natural images seem very different from the domain of geometric objects to be few-shot generalizable.

**Documentation:**

The dataset was easy to access with the url.

**Ethics:**

The authors mention that this dataset/analysis will not lead to any obvious ethical concerns. I agree with this premise.

**Relation To Prior Work:**

Disentanglement literature will greatly benefit from building geometric objects and then understanding the nuances of interpolations of geometric objects. Most likely this wasn’t the author’s motivation but just wanted to highlight this information here.

Locatello, Francesco, et al. "Challenging common assumptions in the unsupervised learning of disentangled representations." international conference on machine learning. PMLR, 2019.

**Summary And Contributions:**

The paper presents a very interesting dataset of Euclidean shapes and objects. It shows the limitations of current vision models in understanding these shapes and geometrical structures. This task is especially important since humans are good at it and current vision models are not. The paper is very well written and it is easy to understand the task and the motivation. The authors also introduce a dataset and a library to generate new datasets using the introduced geometric primitives.

---

> ### Author Response · Authors · 2022-08-14
> **Response to Reviewer BME1 pt 1**
>
> We thank you for the constructive feedback!
>
> Q: Pretrained model on natural images.
>
> A: We thank you for bringing up this point. We would like to clarify that our goal in this paper is to present a task that tests a core component of human vision; humans successfully achieve high accuracy on the Geoclidean tasks even if most exposure is to natural images. Indeed, Kennedy and Ross [1] showed that humans without significant exposure to pictorial art are able to identify a wide range of outline drawings. We thus believe it is an interesting task to evaluate whether pretrained vision models are intrinsically sensitive to Euclidean geometry as humans are. Other works [2] have also shown that high level features of natural images and line drawings are similar for neural visual systems, even on convolutional networks optimized to recognize objects in only photographs but not drawings. However, the Geoclidean DSL and rendering library can also be used for large scale training and fine-tuning, and we are excited to see how the community builds upon it.
>
> Meanwhile, we agree with you that evaluating models pre-trained on line drawings would offer additional perspectives and strengthen the submission. Hence, based on your suggestion, we have added an evaluation of vision models pretrained on the Google Quick Draw dataset [3]. We have trained the VGG16, ResNet50, InceptionV3, and Vision Transformer model on the Google Quick Draw dataset, and tested them on all Geoclidean tasks. On the Quick Draw dataset, these models achieved a top-1 candidate test accuracy of 0.5455, 0.6752, 0.6921, and 0.7039, respectively. These values are in the same range as the reported value of approximately 0.7 for the released recurrent neural network in the official Github repository [4].
>
> Below, we report the average accuracy for Quick Draw–pretrained models across all Geoclidean tasks, for each vision model and for low-level and high-level features, compared to the ImageNet pretrained models. We see that ImageNet pretrained models are comparable with Quick Draw–pretrained models in low-level features, and outperform Quick Draw–pretrained models in high-level features. We hypothesize that though the visual domain shift from Quick Draw to Geoclidean is smaller than that of ImageNet to Geoclidean, there is added complexity of geometric reasoning in our proposed tasks that is not captured by drawing classification. Hence, this illustrates precisely what we aim to test with Geoclidean.
>
> |                       | Human  | VGG16  |      | RN50  |      | INV3  |      | VIT  |      |
> | -----------           | ------ | ------ | -----| ------| -----| ------| -----| ------| -----|
> |                       |        | Low    | High | Low    | High | Low    | High | Low    | High |
> | ImageNet   pretrained | 0.91   | 0.55   | 0.60 | 0.55  | 0.53 | 0.54  | 0.60 | 0.55  | 0.70 |
> | Quick Draw pretrained | 0.91   | 0.52   | 0.54 | 0.55  | 0.53 | 0.56  | 0.55 | 0.55  | 0.57 |
>
> We have added results from Quick Draw pretrained models with accuracy for each task to the Appendix (A.5 Pretraining).
>
> [1] Kennedy, J. M., & Ross, A. S. (1975). Outline picture perception by the Songe of Papua. Perception, 4(4), 391-406.
>
> [2] Fan, J. E., Yamins, D. L., & Turk‐Browne, N. B. (2018). Common object representations for visual production and recognition. Cognitive science, 42(8), 2670-2698.
>
> [3] Ha, D., & Eck, D. (2017). A neural representation of sketch drawings. ICLR 2018.
>
> [4] https://github.com/tensorflow/docs/blob/master/site/en/r1/tutorials/sequences/recurrent_quickdraw.md

---

> > ### Author Response · Authors · 2022-08-14
> > **Response to Reviewer BME1 pt 2**
> >
> > Q: Additional analysis on low-level features.
> >
> > A: We thank you for the recommendation. We have added additional analysis on low-level features to the Appendix (A.4 Feature Visualizations). We study low-level features from ResNet50 and the Vision Transformer on a variety of rendered Geoclidean images. In Figure 2 of the Appendix, we investigate three comparisons of feature space, with PCA dimension reduction applied to each feature to plot in two-dimensional space. The three comparisons are the following: 1) on circles and lines, the most simple geometric primitives, 2) on two different complex Geoclidean concepts which contain different primitives, two circles and one line vs. two lines and one circle, and 3) on positive and negative examples from a Geoclidean task, which contains renderings from related concepts that differ by a few constraint differences with the same primitives.
> >
> > We see that for both ResNet50 and ViT low-level features, the first two tasks involving different geometric structures achieve strong separation; this holds true for both the simpler and complex case, showing that low-level features do generalize with respect to some visual geometric differences. In contrast, the third Geoclidean task is difficult for both vision models, highlighting the intended difficulty of Euclidean geometric reasoning – why our task is especially interesting. We note that low-level features achieve the same accuracy for both ResNet50 and Vision Transformers, with 0.55 accuracy, as seen in Table 3 of the main text.
> >
> > ------
> >
> > Q: Additional analysis on the manifold.
> >
> > A: We have added additional analysis on near-far categorization in the feature space to the Appendix (A.4 Feature Visualizations). We apply PCA dimension reduction on high-level features from ResNet50 and study comparisons across different rendered images, for concepts in both Elements and Constraints dataset. In Figure 3 of the Appendix, we see feature visualizations from six different groups of rendered images for each base concept. The red, orange, green, blue groups are derived from the Geoclidean task, which include positive examples of the target concept (red and orange), as well as negative examples from related concepts (Far as green and Close as blue). The purple and pink groups are simple perturbations of the target train images: scaled (purple) and warped (pink).
> >
> > We can see that ResNet50 features are robust to simple perturbations in geometry, as purple and pink images form distinct clusters for near-far categorization. In contrast, the rendered images from related concepts are difficult to distinguish from the target concept, as red, orange, green, and blue are not disentangle-able. While these vision models do learn near and far categorizations well from the geometric perspective, geometric reasoning is significantly more difficult; hence Geoclidean is an especially interesting task.
> >
> > ------
> >
> > Q: Disentanglement literature.
> >
> > A: We thank you for the recommendation, and agree that disentanglement literature can benefit from Geoclidean's ability to flexibly build geometric objects with different interpolations across factors of variation. We have highlighted this connection and the suggested paper in the Appendix (A.7 Discussion).
> >
> > ------
> >
> > We have highlighted all changes made in the main text and Appendix in red for clarity; we hope we have addressed your concerns, and would be happy to further discuss.

---

### Author Response · Authors · 2022-08-25
**Looking forward to discussion**

Dear Reviewers,

Thank you again for the constructive reviews, which have helped us greatly improve the quality and clarity of our paper. We hope that we have been able to address your concerns. As we approach the end of the discussion period, please don’t hesitate to let us know if you have any additional questions or comments!

Thanks for your time,

Authors

---

### Meta-Review · Area_Chair_3e85 · 2022-09-08

**Recommendation:** Accept
**Confidence:** 5

**Metareview:**

This paper introduces a new dataset to evaluate the discrimination of geometric shapes. Using these images, the paper shows that humans (who probably have had extensive training with geometry) seem to generalize better across geometric shapes than standard computer vision algorithms which have had no direct exposure to training on geometric shapes but have been trained with natural images.

---

### Decision · Program_Chairs · 2022-09-16

Accept